

# Global symmetry and conformal bootstrap in the two-dimensional $O(N)$ model

Linnea Grans-Samuelsson[1★], Rongvoram Nivesvivat[1†], Jesper L. Jacobsen[1,2,3‡],
Sylvain Ribault[1∘] and Hubert Saleur[1,4§]

**1** Institut de physique théorique, CEA, CNRS, Université Paris-Saclay
**2** Laboratoire de Physique de l'École Normale Supérieure, ENS, Université PSL, CNRS,
Sorbonne Université, Université de Paris, F-75005 Paris, France
**3** Sorbonne Université, École Normale Supérieure,
CNRS, Laboratoire de Physique (LPENS), F-75005 Paris, France
**4** Department of Physics and Astronomy, University of Southern California, Los Angeles

★ anna-linnea.grans-samuelsson@ipht.fr , † rongvoramnivesvivat@gmail.com ,
‡ jesper.jacobsen@ens.fr , ∘ sylvain.ribault@ipht.fr , § hubert.saleur@ipht.fr

## Abstract

We define the two-dimensional $O(n)$ conformal field theory as a theory that includes the critical dilute and dense $O(n)$ models as special cases, and depends analytically on the central charge. For generic values of $n \in \mathbb{C}$, we write a conjecture for the decomposition of the spectrum into irreducible representations of $O(n)$. We then explain how to numerically bootstrap arbitrary four-point functions of primary fields in the presence of the global $O(n)$ symmetry. We determine the needed conformal blocks, including logarithmic blocks, including in singular cases. We argue that $O(n)$ representation theory provides upper bounds on the number of solutions of crossing symmetry for any given four-point function. We study some of the simplest correlation functions in detail, and determine a few fusion rules. We count the solutions of crossing symmetry for the 30 simplest four-point functions. The number of solutions varies from 2 to 6, and saturates the bound from $O(n)$ representation theory in 21 out of 30 cases.



# 1  Introduction

In this article we initiate the application of the bootstrap approach to the two-dimensional $O(n)$ conformal field theory, with the specific goals of understanding how the $O(n)$ symmetry acts on the spectrum, and how it manifests itself in crossing symmetry equations. It is in principle enough to define the theory as a set of CFT data, namely a space of states and the corresponding structure constants. However, this raises the issue of making contact with the continuum limit of the $O(n)$ lattice model [1]. We will start with a quick review of the lattice model, before defining the $O(n)$ conformal field theory.

**The two-dimensional $O(n)$ model and its lattice description**

The $O(n)$ model can be defined either on a lattice, or directly as a field theory on a continuous space via a Lagrangian. The lattice description has the advantages of allowing the torus partition function to be computed, and of allowing the model to be defined for non-integer values of $n$. These features are crucial to the bootstrap investigation that we undertake here, which starts with the torus partition function [2], and numerically solves crossing symmetry equations at complex values of $n$. The analyticity in $n$ is less clear in the Lagrangian description, as we will discuss in Section 5.3.

On each vertex $x$ of a honeycomb lattice, let $\phi(x)$ be a variable with $n$ components, subject to the quadratic constraint $\phi(x) \cdot \phi(x) = 1$, and transforming in the vector representation of the group $O(n)$ [1, 3]. Each variable interacts with its nearest neighbors, and the weight of a configuration is $w(\{\phi(x)\}) = \prod_{<x,y>} (1 + K\phi(x) \cdot \phi(y))$, where $<x, y>$ denote nearest neighbor pairs. Instead of a sum over configurations of $\phi(x)$, the partition function can be reformulated in a high-temperature (small $K$) expansion as a sum over configurations of loops where each edge is occupied at most once. The contribution of a configuration comes with a factor $K$ for each occupied edge, and a factor $n$ for each closed loop. In this formulation, the model is called a dilute loop gas.

For any $-2 \le n \le 2$, there is a critical value [1]

$$K_c(n) = \frac{1}{\sqrt{2 + \sqrt{2 - n}}} . \tag{1.1}$$

(For $n = 0$, this formula has been proved rigorously [4].) The model has four phases:

- For $0 < K < K_c(n)$, a high-temperature massive phase.

- At $K = K_c(n)$, the continuum limit of the critical point is a CFT, which we call the **critical dilute $O(n)$ model**.

- For $K_c(n) < K < \infty$ and $-2 < n < 2$, there is also a critical phase, called the dense loop gas. Its properties do not depend on the value of $K$. The continuum limit of that phase is called the **critical dense $O(n)$ model** [5].

- For $K = \infty$ and $-2 < n \leq 2$ the model exhibits a distinct critical phase, the fully-packed loop (FPL) gas in which the loops jointly cover all the lattice vertices [6]. This phase and its corresponding CFT are specific to the honeycomb lattice [7] and we shall not consider them further.

The cases $n = \pm 2$ are a bit special. For $n = -2$ and $K > K_c(n)$ the lattice model experiences a first-order phase transition. For $n = 2$ the continuum limits of the dilute and dense phases coincide. Other versions of the lattice model exist:

- The same model can be put on a square lattice. There is ample analytic and numerical evidence that this does not change the critical phases and their CFT description [8]. However, the square-lattice model also allows for a richer choice of multicritical interactions, which lead to extra critical phases [9–11].

- On the honeycomb lattice, we can use the alternative configuration weight $w(\{\phi(x)\}) = \prod_{<x,y>} \exp(K\phi(x) \cdot \phi(y))$. It is widely believed that this does not change the phase diagram, and leads to the same critical dilute $O(n)$ model at $K = K_c(n)$ [8,12]. However, the limit $K \to \infty$ becomes ill-defined, and the continuum limit of the critical phase at $K > K_c(n)$ can change. This same change also occurs on the square lattice, where it is due to four-leg crossings becoming relevant [12].

In the loop gas description of the lattice model, $n$ needs no longer be integer, and appears as a continuous parameter. On a finite lattice, correlation functions (including the partition function) are sums over finite numbers of graphs with polynomial $n$-dependent coefficients. After taking the continuum limit, $n$ is still a continuous parameter of the resulting CFT. This is supposed to hold not only for the dilute loop gas, but also for the dense loop gas, at least if $n \neq 0$. There are however subtleties: in the dense loop gas, it is known that the limit $n \to 0$ does not commute with the continuum limit [13].

What we lose in the loop gas description is unitarity. Whether two given occupied edges belong to the same loop is a non-local question, and this non-locality can be avoided by the introduction of local, complex weights [8]. This leads to the expectation that the continuum limit of the loop model is a non-unitary CFT. This is true even if $n$ is integer. For example, the $O(1)$ model coincides with the Ising model, whose local observables are described by a unitary minimal model. However, the loop model allows us to define non-local observables, whose critical limits do not belong to the minimal model, but to a larger, non-unitary CFT.

**The $O(n)$ conformal field theory**

As two-dimensional CFTs, the critical dilute and dense $O(n)$ models are characterized by their central charges, which are functions of $n$. These functions are better expressed via a parameter $\beta^2$:

$$c = 13 - 6\beta^2 - 6\beta^{-2}, \quad n = -2\cos(\pi\beta^2). \tag{1.2}$$

Then the difference between the dense and dilute models is the range of values of $\beta^2$:

| Critical model | $n$ | $\beta^2$ | $c$ |
|---|---|---|---|
| Dilute | $[-2, 2]$ | $[1, 2]$ | $[-2, 1]$ |
| Dense | $(-2, 2)$ | $(0, 1)$ | $(-\infty, 1)$ |

(1.3)

In particular, for any $c \in [-2, 1)$, the dilute and dense models are two distinct CFTs with the same central charge. Remarkably, as functions of $\beta^2$, the correlation functions of these two CFTs are given by the same expressions. In principle, this can be understood by reformulating the honeycomb-lattice loop model as a square-lattice loop model in one particular regime, which comprises both the dense and dilute $O(n)$ models in their entire critical ranges $-2 < n \leq 2$ [8]. In practice, this is confirmed by numerical studies [8], including for the three-point functions [14].

We define the $O(n)$ **conformal field theory** as a family of CFTs parametrized by $\beta^2$, which includes the critical dilute and dense $O(n)$ models as special cases. Actually, just like the critical $Q$-state Potts model, the $O(n)$ CFT makes sense way beyond the interval $\beta^2 \in (0, 2)$ that covers these two models: the allowed range of $\beta^2$ is the complex half-plane [15]

$$\Re\beta^2 > 0 \implies \Re c < 13 . \tag{1.4}$$

The $O(n)$ CFT therefore lives on a $\beta^2$-half-plane, or equivalently on a double cover of a $c$-half-plane, or equivalently on a covering of the $n$-complex plane with infinitely many sheets. Let us draw the $\beta^2$-complex plane, where the allowed range is divided into strips of width one. We call the first two strips dense and dilute, by extension of that terminology to complex values of the parameters:

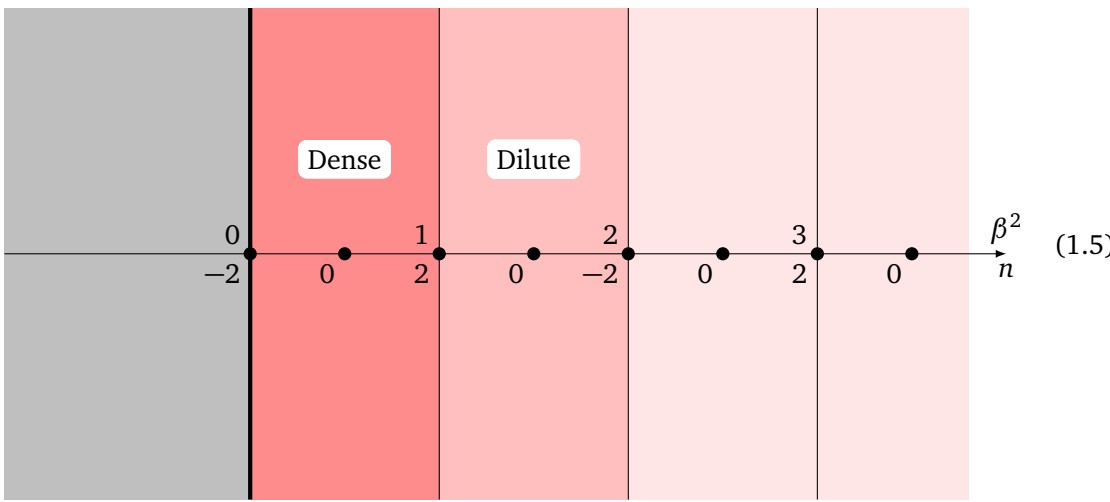

(1.5)

For some integer values of $n$, the $O(n)$ CFT describes models of particular physical interest:

| $n$ | $c_{\text{dilute}}$ | Dilute model | $c_{\text{dense}}$ | Dense model |
|---|---|---|---|---|
| $-2$ | $-2$ | Symplectic fermion (LERW) | $-\infty$ | Not defined |
| $-1$ | $-\frac{3}{5}$ | Related to spanning forests? | $-7$ | Related to spanning forests? |
| $0$ | $0$ | Dilute polymers (SAW) | $-2$ | Dense polymers |
| $1$ | $\frac{1}{2}$ | Ising model | $0$ | Percolation hulls ($T = \infty$ Ising) |
| $2$ | $1$ | Free boson | $1$ | Free boson |

(1.6)

Some comments and references:

- $n = 2$: The relation with the free boson follows directly from the Coulomb gas mapping [1,2].

- $n = 1$ (dilute): The equivalence with the Ising model on a triangular lattice is a standard result of low-temperature expansion of the latter in terms of domain walls on the dual (honeycomb) lattice. Actually, the critical coupling $K_c(1) = \frac{1}{\sqrt{3}}$ matches the known critical coupling of the triangular-lattice $Q$-state Potts model at $Q = 2$ [16].

- $n = 1$ (dense): The infinite-temperature limit of the Ising model on the triangular lattice resides in the dense phase ($O(n)$ loops are the domain walls). It corresponds to the trivial value $K = 1$, which identifies the corresponding loops with the hulls of site-percolation clusters on the triangular lattice.

- $n = 0$: The polymer limits are extensively discussed in [5]. Dilute polymers are also known as self-avoiding walks (SAW) and provide arguably the single most important motivation for studying the $O(n)$ model.

- $n = -1$: The subtle relation between spanning forests and a non-linear sigma model with $n = -1$ components is treated in [17], but the relation to the $O(-1)$ models remains speculative [18].

- $n = -2$: The link to symplectic fermions and the loop-erased random walks (LERW) is covered by [19,20].

**Solving the $O(n)$ CFT: a brief history**

In dimensions between two and four, the critical $O(n)$ model was the subject of early works by Lang and Rühl, who found quite a few nontrivial results on the spectrum and fusion rules [21].

The study of the two-dimensional $O(n)$ CFT has been closely intertwined with the study of the critical $Q$-state Potts model, which is technically very similar. While the central charge and the spectrum of conformal dimensions have been known for a long time [1, 2, 22], the determination of four-point functions and operator product expansions has long remained inaccessible, due to the absence of BPZ differential equations for most correlation functions of the theory. Progress came from work on lattice models and their algebraic aspects [23–26], from considerations of symmetry [27], and from the bootstrap approach [15, 28–32].

However, the bootstrap approach has only been applied to the Potts model so far, and only in the case of four-point connectivities, which are the simplest nontrivial four-point correlation functions. In this article we will apply the bootstrap approach to the $O(n)$ CFT, and start a systematic scan of the model's four-point functions. To do this, we have developed and adapted numerical bootstrap code that was originally written for Liouville theory, the $Q$-state Potts model, and related CFTs [33].

One crucial aspect of our approach is to label primary fields $V_{(r,s)}^{\lambda}$ by both their conformal dimensions, in the form of Kac indices $(r, s)$, and irreducible representations of $O(n)$, in the form of Young tableaux $\lambda$. We consider only generic parameter values $\beta^2 \notin \mathbb{Q}$, in order to keep the structures of indecomposable representations under control.

**Highlights of this article**

Let us point out a few ideas and results that we consider particularly worthy of attention:

- The definition of the $O(n)$ CFT over the complex $\beta^2$-plane, earlier in this introduction.

- The conjectured decomposition of the spectrum into irreducible representations of $O(n)$, Eqs. (2.20) and (2.26).

- The principles of the conformal bootstrap method in the presence of a global symmetry in Section 3.2, in particular the inequality (3.30) between numbers of bootstrap solutions and $O(n)$ invariants.

- The fusion rules of the fields $V_{(\frac{1}{2},0)}^{[1]}$, $V_{(1,0)}^{[2]}$ and $V_{(1,1)}^{[11]}$, in Sections 4.1 and 4.2.

## 2 The group $O(n)$ and its action on the spectrum

In order to solve the model, we need to determine its spectrum, i.e., its space of states. The known symmetries of the model are conformal symmetry, which is described by a product $\mathfrak{C}$ of two Virasoro algebras (left-moving and right-moving), and the global $O(n)$ symmetry. The spectrum should therefore decompose into representations of $O(n) \times \mathfrak{C}$.

In the $O(n)$ Wess–Zumino–Witten model, we would have the symmetry $O(n) \times \mathfrak{C}$, which would however be part of a larger symmetry, due to the presence of two conserved $O(n)$ currents: primary fields with conformal dimensions $(\Delta, \bar{\Delta}) = (1, 0)$ and $(0, 1)$, which belong to the adjoint representation of $O(n)$. In the $O(n)$ CFT, we also have two $O(n)$ currents, but they are not independently conserved, have logarithmic OPEs, and do not give rise to a Kac–Moody symmetry algebra.

### 2.1 Partition function and action of the conformal algebra

Our main source of information on the spectrum is the torus partition function. By definition, the partition function counts the generalized eigenvectors of the zero-mode generators $L_0, \bar{L}_0$ of the two Virasoro algebras. This is in principle not enough for decomposing the spectrum into representations of $\mathfrak{C}$, let alone $O(n)$. Nevertheless, that goal can be reached with the help of other sources of information, and of some guesswork.

**Partition function**

The conformal dimensions of the primary states in the $O(n)$ CFT are of the type

$$\Delta_{(r,s)} = P_{(r,s)}^2 - P_{(1,1)}^2 \quad \text{with} \quad P_{(r,s)} = \frac{1}{2}\left(\beta r - \beta^{-1}s\right), \tag{2.1}$$

where the Kac table indices $r, s$ take values in $\mathbb{Q}$. The relevant characters of the conformal algebra are the diagonal degenerate characters

$$\chi_{\langle r,s\rangle}(q) = \left|\frac{q^{P_{(r,s)}^2} - q^{P_{(r,-s)}^2}}{\eta(q)}\right|^2, \tag{2.2}$$

as well as the non-diagonal characters

$$\chi_{(r,s)}^N(q) = \frac{q^{P_{(r,s)}^2}\bar{q}^{P_{(r,-s)}^2}}{|\eta(q)|^2}. \tag{2.3}$$

In these expressions, $q = e^{2\pi i\tau}$ is the exponentiated modulus of the torus, and $\eta(q)$ is the Dedekind eta function.

The partition function was obtained as early as 1987 by calculating the continuum limit of the lattice partition function on the torus [2]. The lattice partition function is a sum over all

loop configurations on a doubly periodic system. By construction, the lattice partition function is already modular invariant. Its continuum limit is

$$Z^{O(n)}(q) = \sum_{s \in 2\mathbb{N}+1} \chi_{\langle 1,s \rangle}(q) + \sum_{r \in \frac{1}{2}\mathbb{N}^*} \sum_{s \in \frac{1}{r}\mathbb{Z}} L_{(r,s)}(n) \chi_{(r,s)}^N(q) , \qquad (2.4)$$

where $L_{(r,s)}(n)$ is a polynomial function of $n$ that we will write in Eq. (2.15). For the moment, we will focus on the dependence on $q$, and what it reveals on the representations of the conformal algebra that appear in the spectrum.

The partition function is a sum over Kac indices $r,s$. In a minimal model, these indices would take finitely many integer values. In the $O(n)$ CFT, the spectrum is much richer. Both indices can take infinitely many values, and the second index can take fractional values with arbitrarily high denominators, provided the conformal spin $rs$ remains integer. Faced with such a rich spectrum, we may be tempted to look for a larger symmetry algebra that would help organize it. However, the presence of arbitrarily high denominators dooms such ideas, and indeed it is known that (except for $n = 1, 2$) there exists no chiral algebra that would organize the spectrum into finitely many representations, in other words that would make the CFT rational [23]. Actually, the CFT is not even quasi-rational, i.e. the fusion product of two representations may include infinitely many indecomposable representations.

The integer values of the indices that do appear in the $O(n)$ CFT lead to algebraic complications. For $r,s \in \mathbb{N}^*$, a primary field of dimension $\Delta_{(r,s)}$ has a null vector. In a unitary CFT, null vectors would have to vanish. In the $O(n)$ CFT, null vectors do not necessarily vanish, and they lead to the existence of logarithmic representations. Before reviewing these representations for generic values of $\beta^2$, let us point out that the situation is even more complicated if $\beta^2 \in \mathbb{Q}$: in this case, more null vectors appear, leading to more intricate algebraic structures that are just beginning to be understood [34].

**Logarithmic representations of the conformal algebra**

Let us briefly review the action of the conformal algebra on the spectrum of the $O(n)$ CFT, which was recently determined in [26, 32]. (For earlier partial results, see [24, 27].)

The appearance of degenerate characters in the partition function strongly suggests that the corresponding degenerate representations $\mathcal{R}_{\langle 1,s \rangle}$ appear in the spectrum, and we will work under that assumption. To be precise, $\mathcal{R}_{\langle 1,s \rangle}$ is the tensor product of the degenerate representation of the left-moving Virasoro algebra with a vanishing null vector at level $s$, with the same degenerate representation of the right-moving Virasoro algebra.

In the non-diagonal sector, the character $\chi_{(r,s)}^N(q)$ can only describe a Verma module unless $r,s \in \mathbb{Z}^*$. In that case, due to the existence of null vectors in the Verma module, there exist other representations with the same character, including an infinite family of logarithmic representations. To lift this ambiguity, one approach is to relate the ambiguous case $(r,s) \in \mathbb{Z}^*$ to the unambiguous case $(r,0)$ via fusion with degenerate fields [24, 27, 32]. Another approach is to take the conformal limit of the lattice model [26]. Both approaches converge on the same results, i.e., on an indecomposable logarithmic representation whose character is $\chi_{(r,s)}^N(q) + \chi_{(r,-s)}^N(q)$. We introduce the notations

$$\mathcal{W}_{(r,s)} \underset{r,s \in \mathbb{N}^*}{=} \text{indecomposable representation with character } \chi_{(r,s)}^N(q) + \chi_{(r,-s)}^N(q) , \qquad (2.5a)$$

$$\mathcal{W}_{(r,s)} \underset{r,-s \in \mathbb{N}^*}{=} 0 , \qquad (2.5b)$$

$$\mathcal{W}_{(r,s)} \underset{r \notin \mathbb{Z}^* \text{ or } s \notin \mathbb{Z}^*}{=} \text{Verma module with character } \chi_{(r,s)}^N(q) . \qquad (2.5c)$$

The convention of setting some representations to zero is meant to avoid overcounting in the spectrum (2.20), where we will have combinations of the type $\mathcal{W}_{(r,s)} \oplus \mathcal{W}_{(r,-s)}$ whenever $rs \neq 0$.

We refrain from reviewing the structures of the logarithmic representations in more detail, as we will not need them. What we do need are the corresponding conformal blocks, which we will discuss in Section 3.1.

## 2.2 Representations of $O(n)$ and their tensor products

Much is known about finite-dimensional representations of $O(n)$, whether $n$ is integer or generic, but it is not always easy to find the relevant results in the mathematical literature. We will briefly review the results that we need. For more information and references, see the Wikipedia article on Representations of classical Lie groups, which some of us recently created.

### $O(n)$ symmetry with non-integer $n$

Consider the $n$-dimensional vector representation of $O(n)$. A state in that representation shows up in the model's torus partition function (2.4) as a term $Z^{O(n)}(q) = n\chi^{N}_{(\frac{1}{2},0)}(q) + \cdots$. How do we deal with the corresponding fields? At first sight, it seems we must introduce a vector index $i \in \{1, 2, \ldots, n\}$ and write a vector field as $V_i^{[1]}$, where $[1]$ denotes the vector representation. (We omit other parameters such as the conformal dimension.) Then the fusion of this field with itself reads

$$V_{i_1}^{[1]} V_{i_2}^{[1]} \sim \delta_{i_1,i_2} V^{[]} + V_{(i_1,i_2)}^{[2]} + V_{[i_1,i_2]}^{[11]} \,, \tag{2.6}$$

i.e., the $O(n)$ symmetry allows three representations: the singlet representation $[]$, the symmetric traceless tensor $[2]$ and the antisymmetric tensor $[11]$. However, we can make sense of $O(n)$ symmetry with $n$ generic by simply omitting the vector indices, and writing the fusion rule

$$V^{[1]} V^{[1]} \sim V^{[]} + V^{[2]} + V^{[11]} \,. \tag{2.7}$$

Now fields are no longer labelled by states in $O(n)$ representations, but only by the representations themselves. It no longer matters whether these representations have integer dimensions or not. Mathematically, this can be interpreted in terms of tensor categories of representations [35]. In this article, we will sometimes use $O(n)$ vector indices for explanatory purposes, or in order to relate our results to other approaches. However, our results themselves will always hold for generic values of $n$.

It turns out that the generic $n$ case can be obtained formally from the integer $n$ case by taking the limit $n \to \infty$. In our example, the fusion rule is only true if $n \geq 2$, as for $n = 1$ the representations $[2]$ and $[11]$ are actually zero. The fusion rule stabilizes at $n = 2$ and no longer changes as $n$ increases. This stabilization at finite $n$ is a general feature of the tensor products of $O(n)$ representations.

### Irreducible representations and their dimensions

The finite-dimensional irreducible representations of $O(n)$ are parametrized by Young diagrams, which we write as decreasing sequences of natural integers, such as $[53321111] = [53^2 21^4]$. For example, $[k]$ is the fully symmetric representation with $k$ indices, and $[1^k]$ is the fully antisymmetric representation with $k$ indices.

For $\lambda$ a Young diagram, let $\lambda_i$ be the length of the $i$-th row, in other words $\lambda = [\lambda_1 \lambda_2 \cdots \lambda_r]$. Let $\tilde{\lambda}_i$ be the length of the $i$-th column. Let $h_\lambda(i,j) = \lambda_i + \tilde{\lambda}_j - i - j + 1$ be the hook length of

the box $(i, j)$ in the diagram $\lambda$.

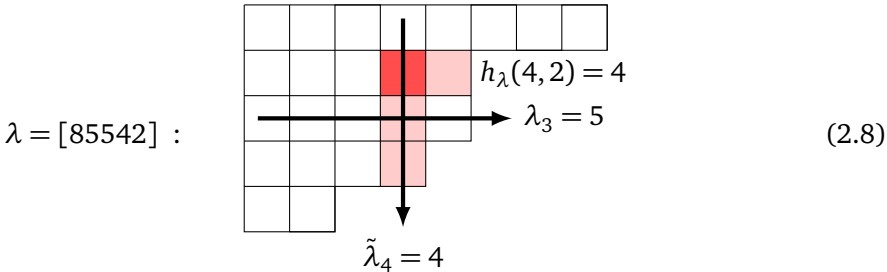

$$\lambda = [85542] : \qquad\qquad\qquad\qquad\qquad\qquad\qquad\qquad\qquad (2.8)$$

Then the dimension of the corresponding $O(n)$ representation is [36]

$$\dim_{O(n)} \lambda = \prod_{\substack{(i,j)\in\lambda \\ i\geq j}} \frac{n + \lambda_i + \lambda_j - i - j}{h_\lambda(i,j)} \prod_{\substack{(i,j)\in\lambda \\ i<j}} \frac{n - \tilde{\lambda}_i - \tilde{\lambda}_j + i + j - 2}{h_\lambda(i,j)} \, . \qquad (2.9)$$

For example, the dimensions of the fully symmetric representations $[], [1], [2], [3], [4], \cdots$ are

$$\dim_{O(n)}[k] = 1, n, \tfrac{1}{2}(n+2)(n-1), \tfrac{1}{6}(n+4)n(n-1), \tfrac{1}{24}(n+6)(n+1)n(n-1), \cdots . \quad (2.10)$$

The dimensions of the fully antisymmetric representations are

$$\dim_{O(n)}[1^k] = \binom{n}{k} \, , \qquad\qquad\qquad (2.11)$$

which vanishes for $k > n$. More generally, for integer $n$, irreducible representations are actually parametrized by diagrams such that $\tilde{\lambda}_1 + \tilde{\lambda}_2 \leq n$. There is no such restriction for generic $n$. In this case, the dimensions of representations should be considered as formal polynomial functions of $n$. The degree of a polynomial is the size of the corresponding Young diagram, i.e., the number of boxes.

**Tensor products and Newell–Littlewood numbers**

Tensor products of $O(n)$ representations with $n$ generic can be written as

$$\lambda \otimes \mu = \sum_\nu N_{\lambda,\mu,\nu}\, \nu \, , \qquad\qquad\qquad (2.12)$$

where the tensor product coefficients $N_{\lambda,\mu,\nu}$ are called Newell–Littlewood numbers [37]. These numbers are $n$-independent natural integers. They can in principle be obtained as large $n$ limits of their integer-$n$ counterparts, although the integer-$n$ coefficients are actually more complicated. For example,

$$[1] \otimes [1] = [2] + [11] + [] \, , \qquad\qquad\qquad (2.13a)$$
$$[1] \otimes [2] = [21] + [3] + [1] \, , \qquad\qquad\qquad (2.13b)$$
$$[1] \otimes [11] = [111] + [21] + [1] \, , \qquad\qquad\qquad (2.13c)$$
$$[1] \otimes [21] = [31] + [22] + [211] + [2] + [11] \, , \qquad\qquad\qquad (2.13d)$$
$$[1] \otimes [3] = [4] + [31] + [2] \, , \qquad\qquad\qquad (2.13e)$$
$$[2] \otimes [2] = [4] + [31] + [22] + [2] + [11] + [] \, , \qquad\qquad\qquad (2.13f)$$
$$[2] \otimes [11] = [31] + [211] + [2] + [11] \, , \qquad\qquad\qquad (2.13g)$$
$$[11] \otimes [11] = [1111] + [211] + [22] + [2] + [11] + [] \, , \qquad\qquad\qquad (2.13h)$$

$$[21] \otimes [3] = [321] + [411] + [42] + [51] + [211] + [22] + 2[31] + [4] + [11] + [2] \, . \tag{2.13i}$$

The tensor product is commutative and associative, and $N_{\lambda,\mu,\nu}$ is symmetric under permutations of the three Young diagrams. The size $|\lambda| = \sum_i \lambda_i$ is conserved modulo 2 and obeys the inequalities

$$||\lambda| - |\mu|| \leq |\nu| \leq |\lambda| + |\mu| \, . \tag{2.14}$$

In practice, all tensor products can be computed using associativity, together with the Pieri-type rule that determines the products of the type $[k] \otimes \mu$. The rule says that $[k] \otimes \mu$ is the sum of all possible Young diagrams that are obtained by, for each successive $i \in \{0, \ldots, k\}$, first removing $i$ boxes from $\mu$ in different columns, and then adding $k - i$ boxes in different columns.

### 2.3 Action of $O(n)$ on the spectrum

**Dimensions of representations**

In the partition function (2.4), the non-diagonal Virasoro characters come with the coefficients [23]

$$\boxed{L_{(r,s)}(n) = \delta_{r,1}\delta_{s\in 2\mathbb{Z}+1} + \frac{1}{2r}\sum_{r'=0}^{2r-1} e^{\pi i r' s} x_{(2r)\wedge r'}(n)} \, , \tag{2.15}$$

where we recall the condition $2r \in \mathbb{N}^*$, and introduce the polynomials $x_d(n)$ such that

$$x_0(n) = 2 \, , \quad x_1(n) = n \, , \quad nx_d(n) = x_{d-1}(n) + x_{d+1}(n) \, . \tag{2.16}$$

If we had set $x_0(n) = 1$, we would have obtained the Chebyshev polynomials of the second kind. Instead, we obtain the polynomials

$$x_2(n) = n^2 - 2 \, , \tag{2.17a}$$

$$x_3(n) = n(n^2 - 3) \, , \tag{2.17b}$$

$$x_4(n) = n^4 - 4n^2 + 2 \, , \tag{2.17c}$$

$$x_5(n) = n(n^4 - 5n^2 + 5) \, , \tag{2.17d}$$

$$x_6(n) = (n^2 - 2)(n^4 - 4n^2 + 1) \, . \tag{2.17e}$$

This leads to the coefficients

$$L_{(\frac{1}{2},0)}(n) = n \, , \tag{2.18a}$$

$$L_{(1,0)}(n) = \tfrac{1}{2}(n+2)(n-1) \, , \tag{2.18b}$$

$$L_{(1,1)}(n) = \tfrac{1}{2}n(n-1) \, , \tag{2.18c}$$

$$L_{(\frac{3}{2},0)}(n) = \tfrac{1}{3}n(n^2-1) \, , \tag{2.18d}$$

$$L_{(\frac{3}{2},\frac{2}{3})}(n) = \tfrac{1}{3}n(n^2-4) \, . \tag{2.18e}$$

By construction, the coefficients obey

$$L_{(r,s)}(n) = L_{(r,-s)}(n) = L_{(r,s+2)}(n) \, . \tag{2.19}$$

These equations are rather easy to interpret. The first equation expresses the invariance of the theory under the exchange of left-moving and right-moving variables. The second equation follows from the existence of a degenerate field with Kac indices $(1, 3)$. We will discuss similar equations for conformal blocks and correlation functions in Section 3.1.

We would now like to write the spectrum of the $O(n)$ CFT as a representation of $O(n) \times \mathfrak{C}$ of the type

$$\mathcal{S}^{O(n)} = \bigoplus_{s \in 2\mathbb{N}+1} [\,] \otimes \mathcal{R}_{\langle 1,s \rangle} \oplus \bigoplus_{r \in \frac{1}{2}\mathbb{N}^*} \bigoplus_{s \in \frac{1}{r}\mathbb{Z}} \Lambda_{(r,s)} \otimes \mathcal{W}_{(r,s)}. \tag{2.20}$$

Here, $\mathcal{R}_{\langle 1,s \rangle}$ and $\mathcal{W}_{(r,s)}$ are the representations of the conformal algebra $\mathfrak{C}$ that we introduced in Section 2.1. The unknown representation $\Lambda_{(r,s)}$ of $O(n)$ is a linear combination of irreducible finite-dimensional representations, with positive integer coefficients that do not depend on $n$ [35, 38]. By definition of the partition function (2.4), this implies

$$\dim_{O(n)} \Lambda_{(r,s)} = L_{(r,s)}(n). \tag{2.21}$$

Given the dimensions (2.9) of $O(n)$ representations, the first three equations of this type have unique solutions,

$$\Lambda_{(\frac{1}{2},0)} = [1], \tag{2.22a}$$

$$\Lambda_{(1,0)} = [2], \tag{2.22b}$$

$$\Lambda_{(1,1)} = [11]. \tag{2.22c}$$

However, the next equation has two solutions, $\Lambda_{(\frac{3}{2},0)} \in \{[3] + [111], [21] + [1]\}$, and the number of solutions increases quickly with $r$. Some extra constraints can be obtained by considering the case $n = 2$ [27], but they are not enough for making the solution unique in general.

**Structures of representations**

In order to write the representations $\Lambda_{(r,s)}$, our basic idea is to use the formula (2.15) for its dimension, where we replace each occurrence of $n$ with an $n$-dimensional formal representation, i.e., a combination of irreducible representations with coefficients in $\mathbb{Z}$. The sizes of the needed Young diagrams are constrained by the requirement that $\Lambda_{(r,s)}$ be a combination of diagrams of size $2r$ or less, since its dimension is a polynomial of degree $2r$. Therefore, for any $t \in \mathbb{N}^*$, we need to find at least one combination of irreducible representations of size $t$ or less, whose dimension is $n$.

We propose the following alternating hook representations,

$$\Lambda_t = \delta_{t \equiv 0 \bmod 2}[\,] + \sum_{k=0}^{t-1} (-1)^k [t-k, 1^k]. \tag{2.23}$$

The first few examples are

$$\Lambda_1 = [1], \tag{2.24a}$$

$$\Lambda_2 = [2] - [11] + [\,], \tag{2.24b}$$

$$\Lambda_3 = [3] - [21] + [111], \tag{2.24c}$$

$$\Lambda_4 = [4] - [31] + [211] - [1111] + [\,]. \tag{2.24d}$$

Automated calculations for many values of $t$ convince us that

$$\dim_{O(n)} \Lambda_t = n . \tag{2.25}$$

This leads us to the conjecture

$$\Lambda_{(r,s)} = \delta_{r,1}\delta_{s\in 2\mathbb{Z}+1}[\,] + \frac{1}{2r}\sum_{r'=0}^{2r-1} e^{\pi i r' s} x_{(2r)\wedge r'}\left(\Lambda_{\frac{2r}{(2r)\wedge r'}}\right) . \tag{2.26}$$

In each polynomial $x_d(n)$ that appears in the formula (2.15) for $L_{(r,s)}(n)$, we have replaced $n$ with a representation $\Lambda_t$ such that $d \times t = 2r$. Replacing powers of $n$ with tensor products of $\Lambda_t$, we obtain the formal representation $x_d(\Lambda_t)$, which is a combination of diagrams of size $2r$ or less. According to the conjecture, the fully symmetric tensor $[2r]$ appears in $\Lambda_{(r,s)}$ with multiplicity $\delta_{s,0}$.

By construction, our representation $\Lambda_{(r,s)}$ has the correct dimension, i.e., Eq. (2.21) is satisfied. However, each term in $\Lambda_{(r,s)}$ is a formal representation, and involves irreducible $O(n)$ representations with coefficients that are neither positive nor even integer. Automated calculations in many examples show that $\Lambda_{(r,s)}$ itself is actually a representation, i.e., a combination with positive integer coefficients. Let us display the first few examples, beyond the cases already given in (2.22a)-(2.22c):

$$\Lambda_{(\frac{3}{2},0)} = [3] + [111] , \tag{2.27a}$$

$$\Lambda_{(\frac{3}{2},\frac{2}{3})} = [21] , \tag{2.27b}$$

$$\Lambda_{(2,0)} = [4] + [22] + [211] + [2] + [\,] , \tag{2.27c}$$

$$\Lambda_{(2,\frac{1}{2})} = [31] + [211] + [11] , \tag{2.27d}$$

$$\Lambda_{(2,1)} = [31] + [22] + [1111] + [2] , \tag{2.27e}$$

$$\Lambda_{(\frac{5}{2},0)} = [5] + [32] + 2[311] + [221] + [11111] + [3] + 2[21] + [111] + [1] , \tag{2.27f}$$

$$\Lambda_{(\frac{5}{2},\frac{2}{5})} = [41] + [32] + [311] + [221] + [2111] + [3] + 2[21] + [111] + [1] , \tag{2.27g}$$

$$\begin{aligned}\Lambda_{(3,0)} = &[6] + 2[42] + 2[411] + [33] + 2[321] + 2[3111] + 2[222] + [2211]\\ &+ [21111] + 2[4] + 4[31] + 4[22] + 4[211] + 2[1111] + 4[2] + 2[11]\\ &+ 2[\,] ,\end{aligned} \tag{2.27h}$$

$$\begin{aligned}\Lambda_{(3,\frac{1}{3})} = &[51] + [42] + 2[411] + [33] + 3[321] + [3111] + 2[2211] + [21111]\\ &+ [4] + 5[31] + 2[22] + 5[211] + [1111] + 2[2] + 4[11] ,\end{aligned} \tag{2.27i}$$

$$\begin{aligned}\Lambda_{(3,\frac{2}{3})} = &[51] + 2[42] + [411] + 3[321] + 2[3111] + [222] + [2211] + [21111]\\ &+ 2[4] + 4[31] + 4[22] + 4[211] + 2[1111] + 4[2] + 2[11] + [\,] ,\end{aligned} \tag{2.27j}$$

$$\begin{aligned}\Lambda_{(3,1)} = &[51] + [42] + 2[411] + 2[33] + 2[321] + 2[3111] + [222] + 2[2211]\\ &+ [111111] + [4] + 5[31] + 2[22] + 5[211] + [1111] + 2[2] + 4[11] .\end{aligned} \tag{2.27k}$$

In the case of $\Lambda_{(2,0)}$, our conjecture agrees with already known results [27]. More evidence will come from our bootstrap calculations, since the structure of $\Lambda_{(r,s)}$ leads to predictions on the numbers of solutions of crossing symmetry equations, as we will explain in Section 3.2. Some of us are working on studying the conjecture in more detail, as well as a similar conjecture for the $Q$-state Potts model, but this is outside the scope of the present article.

# 3  Conformal bootstrap

Since we know the representations of the conformal algebra that appear in the $O(n)$ CFT, we can in principle use the semi-analytic bootstrap method of [28], and write crossing symmetry as a system of linear equations for four-point structure constants. However, the $O(n)$ CFT gives rise to new technical and conceptual issues. In particular, the presence of a global $O(n)$ symmetry leads to the existence of large numbers of solutions of crossing symmetry, which we will have to count and to interpret. This is a priori not easy, because crossing symmetry equations know only about the conformal symmetry, and do not directly encode any information about the global symmetry.

Let us introduce notations for primary fields in the $O(n)$ CFT, which correspond to the representations in the spectrum (2.20):

- Let $V^D_{\langle 1,s\rangle}$ be a diagonal degenerate primary field: such fields always transform in the singlet representation $[\,]$ of $O(n)$.

- Let $V_{(r,s)}$ be a non-diagonal primary field with the left and right conformal dimensions $(\Delta, \bar\Delta) = (\Delta_{(r,s)}, \Delta_{(r,-s)})$ (2.1).

- Let $V^\lambda$ be a field that belongs to the irreducible representation $\lambda$ of $O(n)$.

- Let $V^\lambda_{(r,s)}$ be a non-diagonal primary field that also belongs to the irreducible representation $\lambda$ of $O(n)$. For example, the representation $\Lambda_{(\frac{3}{2},0)}$ (2.27a) gives rise to the two fields $V^{[3]}_{(\frac{3}{2},0)}$ and $V^{[111]}_{(\frac{3}{2},0)}$.

Unless $r,s \in \mathbb{Z}^*$, the primary field $V_{(r,s)}$ generates the Verma module $\mathcal{W}_{(r,s)}$ of the conformal algebra. If $r,s \in \mathbb{N}^*$, the logarithmic module $\mathcal{W}_{(r,s)}$ contains two primary fields $V_{(r,s)}, V_{(r,-s)}$, which however do not generate it [32]. Our notation is ambiguous whenever $\Lambda_{(r,s)}$ has non-trivial multiplicities. For example, $\Lambda_{(\frac{5}{2},0)}$ (2.27f) gives rise to two independent fields of the type $V^{[21]}_{(\frac{5}{2},0)}$.

## 3.1  Singularities of conformal blocks

In the $O(n)$ CFT, the existence of degenerate fields $V^D_{\langle 1,s\rangle}$ leads to shift equations for structure constants [24,39]. These equations allow us to combine linear sums of infinitely many conformal blocks into interchiral blocks [31,40], and therefore to reduce the number of unknowns in crossing symmetry equations. Moreover, these equations allow us to determine logarithmic conformal blocks [32]. We will now study the singularities that can appear in these equations, and therefore also in conformal blocks.

Before that, let us comment on a small subtlety. The shift equations are usually derived from the existence of the degenerate field $V^D_{\langle 1,2\rangle}$, whereas the spectrum of the $O(n)$ CFT only contains $V^D_{\langle 1,3\rangle}$. Given the fusion rule $V^D_{\langle 1,2\rangle} \times V^D_{\langle 1,2\rangle} \sim V^D_{\langle 1,3\rangle} + V^D_{\langle 1,1\rangle}$, the $V^D_{\langle 1,3\rangle}$ shift equations must follow from the $V^D_{\langle 1,2\rangle}$ shift equations, but they could conceivably be weaker. However, a closer look shows that they are in fact equivalent: monodromies of third-order BPZ equations, while harder to compute, are in principle no less constraining than those of second-order BPZ equations.

**Regularizing singularities in shift equations**

Let us quickly review the shift equations for non-diagonal fields. (The argument would be the same in the presence of diagonal fields.) We assume that there exists a diagonal degenerate

field $V^D_{\langle 1,2 \rangle}$, whose fusion rule with the non-diagonal field $V_{(r,s)}$ is

$$V^D_{\langle 1,2 \rangle} \times V_{(r,s)} \sim V_{(r,s+1)} + V_{(r,s-1)} \,. \tag{3.1}$$

Crossing symmetry and single-valuedness of the four-point function $\langle V^D_{\langle 1,2 \rangle} \prod_{i=1}^{3} V_{(r_i,s_i)} \rangle$ imply the conditions [39]

$$r_i s_i \in \mathbb{Z} \,, \tag{3.2a}$$

$$r_i \in \frac{1}{2}\mathbb{Z} \,, \tag{3.2b}$$

$$r_1 + r_2 + r_3 \in \mathbb{Z} \,. \tag{3.2c}$$

The first two conditions are obeyed by the spectrum (2.20) of the $O(n)$ CFT. The third condition is a basic constraint on fusion: the first index is conserved modulo integers. This constraint is equivalent to the conservation of $|\lambda|$ modulo 2 in tensor products of $O(n)$ representation, since the model only contains fields $V^\lambda_{(r,s)}$ such that $|\lambda| \equiv 2r \bmod 2$.

However, when it comes to the linear system [39](3.16) that leads to shift equations, only the last two conditions are necessary for a solution to exist. When we encounter singularities in shift equations, we can therefore regularize them by relaxing the integer spin condition (3.2a), i.e., by analytically continuing fields in their second index $s_i$ while keeping $r_i$ fixed.

To be concrete, our shift equations (3.5) and (3.6) will involve ratios of Gamma functions, which may include factors of the type $\rho = \frac{\Gamma(\frac{1}{2}r + \frac{1}{2}\beta^{-2}s)}{\Gamma(\frac{1}{2}r - \frac{1}{2}\beta^{-2}s)}$. In the $O(n)$ CFT, we may need the value of this ratio for $r = s = 0$. This value depends on the way we take the limit, in particular $\lim_{r \to 0} \lim_{s \to 0} \rho = 1$ while $\lim_{s \to 0} \lim_{r \to 0} \rho = -1$. From the analysis of the shift equations' derivation, we have just deduced that the correct limit is the second one.

**Interchiral blocks**

Let us consider a four-point function of non-diagonal primary fields, and its $s$-channel decomposition into conformal blocks:

$$\left\langle \prod_{i=1}^{4} V_{(r_i,s_i)} \right\rangle = \sum_{s \in 2\mathbb{N}+1} D_s \mathcal{G}^D_{\langle 1,s \rangle} + \sum_{r \in \frac{1}{2}\mathbb{N}^*} \sum_{s \in \frac{1}{r}\mathbb{Z}} D_{(r,s)} \mathcal{G}_{(r,s)} \,. \tag{3.3}$$

Here we have summed over the whole spectrum of possible representations (2.20), and introduced the corresponding conformal blocks $\mathcal{G}^D_{\langle 1,s \rangle}$ and $\mathcal{G}_{(r,s)}$ for the channel representations $\mathcal{R}_{\langle 1,s \rangle}$ and $\mathcal{W}_{(r,s)}$ respectively. The coefficients $D_s$ and $D_{(r,s)}$, which unlike the blocks do not depend on the fields' positions, are called four-point structure constants.

Actually, we already know that the sum is not over the whole spectrum. To begin with, $r$ is conserved modulo integers, which implies $\sum_{i=1}^{4} r_i \in \mathbb{Z}$, and eliminates half the terms in our decomposition. Furthermore, degenerate representations obey the fusion rule

$$V^D_{\langle 1,s \rangle} \times V_{(r_1,s_1)} \sim \sum_{j \overset{2}{=} 1-s}^{1+s} V_{(r_1,s_1+j)} \,, \tag{3.4}$$

where the sum runs by increments of two. This severely restricts the degenerate representations that can appear in the decomposition, and actually eliminates them completely unless $(r_1, r_3) = (r_2, r_4)$ and $(s_1, s_3) \equiv (s_2, s_4) \bmod (2,2)$.

Let us now discuss the influence of shift equations on the decomposition into conformal blocks. Shift equations determine how four-point structure constants behave under $s \to s+2$, namely [39]

$$\frac{D_{(r,s+1)}}{D_{(r,s-1)}} = (-)^{2r_2+2r_4+1} \prod_{\epsilon,\eta=\pm} \Gamma\left(\epsilon s \beta^{-2} + \eta r\right)^{-\epsilon} \Gamma\left(\tfrac{1-\eta}{2} + (\epsilon s + \eta)\beta^{-2} - r\right)^{-\epsilon}$$
$$\times \frac{M(P_{(r,s)}, P_1, P_2)}{M(P_{(r,-s)}, \bar{P}_1, \bar{P}_2)} \frac{M(P_{(r,s)}, P_3, P_4)}{M(P_{(r,-s)}, \bar{P}_3, \bar{P}_4)}, \quad (3.5)$$

$$\frac{D_{s+1}}{D_{s-1}} = (-)^{2r_2+2r_4+1} \prod_{\epsilon,\eta=\pm} \Gamma\left(\epsilon s \beta^{-2} - \epsilon\right)^{-\epsilon} \Gamma\left(\tfrac{1-\eta}{2} + (\epsilon s + \eta)\beta^{-2} - \epsilon\right)^{-\epsilon}$$
$$\times \frac{M(P_{(1,s)}, P_1, P_2)}{M(-P_{(1,s)}, \bar{P}_1, \bar{P}_2)} \frac{M(P_{(1,s)}, P_3, P_4)}{M(-P_{(1,s)}, \bar{P}_3, \bar{P}_4)}, \quad (3.6)$$

where we introduced the notations

$$\begin{cases} P_i = P_{(r_i,s_i)} \\ \bar{P}_i = P_{(r_i,-s_i)} \end{cases}, \quad M(P_1, P_2, P_3) = \prod_{\pm,\pm} \Gamma\left(\tfrac{1}{2} - \beta^{-1}P_1 \pm \beta^{-1}P_2 \pm \beta^{-1}P_3\right). \quad (3.7)$$

In the $s$-channel decomposition, it is therefore enough to reduce the second index to an interval of length 2,

$$\left\langle \prod_{i=1}^{4} V_{(r_i,s_i)} \right\rangle = D_{s_0} \mathcal{H}_{s_0} + \sum_{r \in \frac{1}{2}\mathbb{N}^*} \sum_{s \in \frac{1}{r}\mathbb{Z} \cap (-1,1]} D_{(r,s)} \mathcal{H}_{(r,s)}, \quad (3.8)$$

provided we introduce the interchiral blocks

$$\mathcal{H}_{s_0} = \sum_{s \in s_0 + 2\mathbb{N}} \frac{D_s}{D_{s_0}} \mathcal{G}^D_{\langle 1,s \rangle}, \quad \mathcal{H}_{(r,s)} = \sum_{j \in 2\mathbb{N}} \frac{D_{(r,s+j)}}{D_{(r,s)}} \mathcal{G}_{(r,s+j)}, \quad (3.9)$$

where $s_0$ is the smallest index that is allowed by the degenerate fusion rules (3.4), namely

$$s_0 = 1 + \min(|s_1 - s_2|, |s_3 - s_4|) \quad \text{if} \quad \begin{cases} (r_1, r_3) = (r_2, r_4), \\ (s_1, s_3) \equiv (s_2, s_4) \bmod (2,2). \end{cases} \quad (3.10)$$

In the interchiral blocks, the ratios of structure constants are determined by the shift equations. Therefore, just like the conformal blocks themselves, interchiral blocks are universal quantities. Working with interchiral blocks rather than conformal blocks reduces the number of unknown four-point structure constants to be determined in the conformal bootstrap.

**Accidentally non-logarithmic blocks**

Let us discuss the conformal blocks that appear in the decomposition (3.3) of the four-point function $\left\langle \prod_{i=1}^{4} V_{(r_i,s_i)} \right\rangle$. Conformal blocks are functions of the channel representation, which can be a Verma module, a degenerate representation, or a logarithmic representation. In all cases, the blocks can be assembled from the well-known Virasoro conformal blocks $\mathcal{F}_\Delta$, which correspond to Verma modules of the left-moving Virasoro algebra, together with their counterparts $\bar{\mathcal{F}}_\Delta$, which correspond to Verma modules of the right-moving Virasoro algebra.

In the cases of Verma modules and degenerate representations, our conformal blocks factorize, and have the simple expressions

$$\mathcal{G}_{(r,s)} \underset{r \notin \mathbb{Z}^* \text{ or } s \notin \mathbb{Z}^*}{=} \mathcal{F}_{\Delta_{(r,s)}} \bar{\mathcal{F}}_{\Delta_{(r,-s)}} , \tag{3.11}$$

$$\mathcal{G}^D_{\langle 1,s \rangle} = \mathcal{F}_{\Delta_{(1,s)}} \bar{\mathcal{F}}_{\Delta_{(1,s)}} . \tag{3.12}$$

For $r,s \in \mathbb{N}^*$, the logarithmic representation $\mathcal{W}_{(r,s)}$ generically gives rise to a logarithmic block. To write this block, let us introduce the behaviour of $\mathcal{F}_\Delta$ near one of its poles,

$$\mathcal{F}_{\Delta_{(r,s)}+\epsilon} = \frac{R_{r,s}}{\epsilon} \mathcal{F}_{\Delta_{(r,-s)}} + \mathcal{F}^{\text{reg}}_{\Delta_{(r,s)}} + O(\epsilon) . \tag{3.13}$$

Here $R_{r,s}$ is called a conformal block residue, and the regularized block $\mathcal{F}^{\text{reg}}_{\Delta_{(r,s)}}$ is generically logarithmic. Reproducing the formula [32](3.43) while slightly changing the notations and the overall normalization, we have

$$\mathcal{G}_{(r,s)} \underset{r,s \in \mathbb{N}^*}{=} \mathcal{F}^{\text{reg}}_{\Delta_{(r,s)}} \bar{\mathcal{F}}_{\Delta_{(r,-s)}} + \frac{R_{r,s}}{\bar{R}_{r,s}} \mathcal{F}_{\Delta_{(r,-s)}} \bar{\mathcal{F}}^{\text{reg}}_{\Delta_{(r,s)}}$$
$$- R_{r,s} \frac{P_{(r,-s)}}{P_{(r,s)}} \left( \mathcal{F}_{\Delta_{(r,-s)}} \bar{\mathcal{F}}_{\Delta_{(r,-s)}} \right)' - R_{r,s} \frac{\ell_{(r,s)}}{2 P_{(r,s)}} \mathcal{F}_{\Delta_{(r,-s)}} \bar{\mathcal{F}}_{\Delta_{(r,-s)}} , \tag{3.14}$$

where the prime is a derivative with respect to the conformal dimension.

Our formula for logarithmic conformal blocks becomes singular in special cases where $R_{r,s} = 0$, which is equivalent to $\bar{R}_{r,s} = 0$. Remembering that the formula was deduced from shift equations, we know that the singularity should be regularized by continuing the blocks in the second indices $s_i$, while keeping $r_i$ fixed: this allows us to compute the ratio of residues $\frac{R_{r,s}}{\bar{R}_{r,s}}$. Moreover, simplifications occur in the formula (3.14): the regularized block (3.13) is no longer logarithmic or even regularized, i.e., $\mathcal{F}^{\text{reg}}_{\Delta_{(r,s)}} = \mathcal{F}_{\Delta_{(r,s)}}$, and the whole second line vanishes. (The coefficient $\ell_{(r,s)}$ has a finite limit.) We are left with

$$\boxed{\mathcal{G}_{(r,s)} \underset{\substack{r,s \in \mathbb{N}^* \\ R_{r,s}=0}}{=} \mathcal{F}_{\Delta_{(r,s)}} \bar{\mathcal{F}}_{\Delta_{(r,-s)}} + \frac{R_{r,s}}{\bar{R}_{r,s}} \mathcal{F}_{\Delta_{(r,-s)}} \bar{\mathcal{F}}_{\Delta_{(r,s)}}} . \tag{3.15}$$

Of course, the representation $\mathcal{W}_{(r,s)}$ itself is still logarithmic in this case, as its structure does not depend on the four-point function we are considering. The disappearance of logarithmic terms in the conformal block is because the logarithmic fields in the representation happen to give vanishing contributions to this particular block. The block is now a linear combination of two Verma module blocks (3.11), with a relative coefficient that is still determined by the structure of $\mathcal{W}_{(r,s)}$.

## 3.2 Global symmetry and crossing symmetry

The four-point functions of the $O(n)$ conformal field theory are subject to two apparently independent types of constraints: conformal symmetry leads to crossing symmetry equations, while $O(n)$ symmetry leads to other constraints. Since the two types of constraints apply to the same four-point functions, they should lead to compatible results, and this will allow us to test our conjecture (2.26) for the action of the $O(n)$ symmetry on the theory's spectrum. This reasoning should apply not only to the $O(n)$ CFT, but also to any CFT with global symmetries.

**Crossing symmetry**

Crossing symmetry is the equality between three decompositions of the same four-point function $\left\langle \prod_{i=1}^4 V_{(r_i,s_i)} \right\rangle$:

$$\sum_{V\in\mathcal{S}^{(s)}} D_V^{(s)} \quad {}^2\!\!>\!\!-\!\!V\!\!-\!\!<^3_{1\;\;\;\;4} \quad = \sum_{V\in\mathcal{S}^{(t)}} D_V^{(t)} \quad {}^2\diagdown\!/^3_{1}\;V\;{}_4 \quad = \sum_{V\in\mathcal{S}^{(u)}} D_V^{(u)} \quad {}^2\;V\;{}^3_{1\;\;\;4} \tag{3.16}$$

$$s\text{-channel} \qquad\qquad t\text{-channel} \qquad\qquad u\text{-channel}$$

In these equations, the known quantities are the spectra $\mathcal{S}^{(s)}, \mathcal{S}^{(t)}, \mathcal{S}^{(u)}$, and the diagramatically represented interchiral blocks. The $s$-channel decomposition is just another notation for the decomposition (3.8). We do not use the condition that four-point structure constants are products of three-point structure constants, and we view crossing symmetry as a system of linear equations whose unknowns are the four-point structure constants $D_V^{(s)}, D_V^{(t)}, D_V^{(u)}$.

In practice, the spectra $\mathcal{S}^{(s)}, \mathcal{S}^{(t)}, \mathcal{S}^{(u)}$ are subsets of the full spectrum of the $O(n)$ CFT. These subsets are determined by the constraints (3.2c) and (3.4), which we now express as conformal fusion rules for our non-diagonal fields:

$$V_{(r_1,s_1)} \times V_{(r_2,s_2)} \sim \delta_{r_1,r_2} \delta_{s_1-s_2 \in 2\mathbb{Z}} \sum_{s\in|s_1-s_2|+1+2\mathbb{N}} V_{\langle 1,s\rangle}^D + \sum_{r\in\frac{1}{2}\mathbb{N}^*\cap(\mathbb{Z}+r_1+r_2)} \sum_{s\in\frac{\mathbb{Z}}{r}} V_{(r,s)} . \tag{3.17}$$

For example,

$$V_{\left(\frac{1}{2},0\right)} \times V_{\left(\frac{1}{2},0\right)} \sim \sum_{s\in 1+2\mathbb{N}} V_{\langle 1,s\rangle}^D + \sum_{r\in\mathbb{N}^*} \sum_{s\in\frac{\mathbb{Z}}{r}} V_{(r,s)} . \tag{3.18}$$

In these fusion rules, we only write primary fields on the right-hand side. In the corresponding OPEs, $V_{(r,s)}$ comes with all its descendant fields. Moreover, if $r,s\in\mathbb{Z}^*$, there also appear other fields from the indecomposable representation $\mathcal{W}_{(r,s)}$, which are not descendants of $V_{(r,s)}$.

According to the fusion rules, our four-point function is non-vanishing provided $\sum_{i=1}^4 r_i \in \mathbb{Z}$. This is the condition for the spectra $\mathcal{S}^{(s)}, \mathcal{S}^{(t)}, \mathcal{S}^{(u)}$ to be non-empty, in which case they are actually infinite. We therefore have infinitely many unknown four-point structure constants. We also have infinitely many equations, since interchiral blocks are functions of one complex variable (the cross-ratio of the four fields' positions). Let us write the number of independent solutions as

$$\boxed{\mathcal{N}_{\left\langle \prod_{i=1}^4 V_{(r_i,s_i)} \right\rangle} = \dim\{\text{solutions of (3.16)}\}} . \tag{3.19}$$

From our experience with the $Q$-state Potts model, we expect that this number is finite [31,32]. In the case of cluster connectivities in the $Q$-state Potts model, the number of solutions is 4. Notice however that crossing symmetry equations that involve only 2 channels out of 3 can have infinitely many solutions. Leaving the third channel unconstrained can lead to spurious solutions and/or to solutions that belong to other CFTs.

**Four-point $O(n)$ invariants**

Let us now consider a four-point function $\left\langle \prod_{i=1}^4 V^{\lambda_i} \right\rangle$ from the point of view of $O(n)$ symmetry. The values of that four-point function are by definition invariant under $O(n)$, so the four-point function may be viewed as an invariant tensor in $\otimes_{i=1}^4 \lambda_i$.

In order to discuss $O(n)$ invariants, let us introduce the notation $\mathrm{Hom}(\Lambda_1, \Lambda_2)$ for the space of intertwiners between two representation $\Lambda_1, \Lambda_2$, i.e. the space of linear maps that commute with the action of $O(n)$. For two irreducible representations, Schur's lemma can be written as $\dim \mathrm{Hom}(\lambda, \mu) = \delta_{\lambda, \mu}$. Moreover, we have canonical isomorphisms $\mathrm{Hom}(\Lambda_1, \Lambda_2) \simeq \mathrm{Hom}(\Lambda_2, \Lambda_1) \simeq \mathrm{Hom}(\Lambda_1 \otimes \Lambda_2, [\,])$, where the latter space is the space of invariant tensors in $\Lambda_1 \otimes \Lambda_2$, i.e. the space of intertwiners between $\Lambda_1 \otimes \Lambda_2$ and the identity representation $[\,]$. The decomposition of a tensor product into irreducible representations can be written as

$$\lambda \otimes \mu = \bigoplus_{\nu} N_{\lambda, \mu, \nu} \nu \quad \text{with} \quad N_{\lambda, \mu, \nu} = \dim \mathrm{Hom}(\lambda \otimes \mu, \nu) \in \mathbb{N} . \tag{3.20}$$

Using this notation, we can write

$$\left\langle \prod_{i=1}^{4} V^{\lambda_i} \right\rangle \in \mathrm{Hom}\left( \bigotimes_{i=1}^{4} \lambda_i, [\,] \right) . \tag{3.21}$$

The dimension of this space, which is also the number of linearly independent $O(n)$ invariants in the representation $\otimes_{i=1}^{4} \lambda_i$, will be denoted as

$$\mathcal{I}_{\left\langle \prod_{i=1}^{4} V^{\lambda_i} \right\rangle} = \dim \mathrm{Hom}\left( \bigotimes_{i=1}^{4} \lambda_i, [\,] \right) . \tag{3.22}$$

Each channel $s$, $t$ or $u$ gives rise to a different basis of invariants, and to a different calculation of this dimension. Consider for example the $s$-channel, and consider the following decompositions into irreducible $O(n)$ representations $\nu$:

$$\lambda_1 \otimes \lambda_2 = \bigoplus_{\nu} N_{\lambda_1, \lambda_2, \nu} \nu, \quad \lambda_3 \otimes \lambda_4 = \bigoplus_{\nu} N_{\lambda_3, \lambda_4, \nu} \nu . \tag{3.23}$$

For each irreducible representation $\nu$ that appears in both $\lambda_1 \otimes \lambda_2$ and $\lambda_3 \otimes \lambda_4$, we can build $N_{\lambda_1, \lambda_2, \nu} N_{\lambda_3, \lambda_4, \nu}$ invariants such that $\nu$ propagates in the $s$-channel, using the intertwiners that underlie our decompositions of $\lambda_1 \otimes \lambda_2$ and $\lambda_3 \otimes \lambda_4$:

$$\mathrm{Hom}\left( \bigotimes_{i=1}^{4} \lambda_i, [\,] \right) \simeq \mathrm{Hom}\left( \lambda_1 \otimes \lambda_2, \lambda_3 \otimes \lambda_4 \right) \tag{3.24}$$

$$\simeq \bigoplus_{\nu} \mathrm{Hom}\left( \lambda_1 \otimes \lambda_2, \nu \right) \otimes \mathrm{Hom}\left( \nu, \lambda_3 \otimes \lambda_4 \right) . \tag{3.25}$$

At the level of dimensions, these isomorphisms lead to the expression

$$\mathcal{I}_{\left\langle \prod_{i=1}^{4} V^{\lambda_i} \right\rangle} = \sum_{\nu} N_{\lambda_1, \lambda_2, \nu} N_{\lambda_3, \lambda_4, \nu} . \tag{3.26}$$

Whenever $N_{\lambda_1, \lambda_2, \nu} N_{\lambda_3, \lambda_4, \nu} = 1$, we call $T_{\nu}^{(s)}$ the unique (up to rescaling) four-point invariant such that $\nu$ propagates in the $s$-channel. If all relevant fusion multiplicities $N_{\lambda_i, \lambda_j, \nu}$ are one, the three bases of invariants can be written as

$$\mathrm{Hom}\left( \bigotimes_{i=1}^{4} \lambda_i, [\,] \right) = \mathrm{Span}\left\{ T_{\nu}^{(s)} \right\}_{\nu} = \mathrm{Span}\left\{ T_{\nu}^{(t)} \right\}_{\nu} = \mathrm{Span}\left\{ T_{\nu}^{(u)} \right\}_{\nu} . \tag{3.27}$$

Let us give two examples, one with trivial multiplicities, the other one with nontrivial multiplicities:

- Case of $\left\langle V^{[1]}V^{[1]}V^{[1]}V^{[1]}\right\rangle$: the $s$-channel basis is made of the three invariants $T^{(s)}_{[]}$, $T^{(s)}_{[2]}$, $T^{(s)}_{[11]}$. If $n$ is integer, we can restore the indices $i \in \{1, 2, \ldots, n\}$ in our four-point function $\left\langle \prod_{j=1}^{4} V^{[1]}_{i_j} \right\rangle$. Using the fusion rule (2.6), we then write the invariants as explicit tensors such as $T^{(s)}_{[]} = \delta_{i_1,i_2}\delta_{i_3,i_4}$.

- Case of $\left\langle V^{[21]}V^{[21]}V^{[1]}V^{[1]}\right\rangle$: in the $t$-channel, $[1] \otimes [21]$ (2.13d) is a sum of 5 irreducible representations with trivial multiplicities, therefore $\mathcal{I}_{\left\langle V^{[21]}V^{[21]}V^{[1]}V^{[1]}\right\rangle} = 5$. In the $s$-channel, the same result follows from $[1] \otimes [1]$ (2.13a) together with

$$[21] \otimes [21] = \Big(\text{Representations with 6 or 4 boxes}\Big) + 2[2] + 2[11] + [] \,. \tag{3.28}$$

**Four-point functions**

Let us decompose a four-point function in the $O(n)$ conformal field theory over a basis $\{T_k\}_k$ of four-point invariants:

$$\left\langle \prod_{i=1}^{4} V^{\lambda_i}_{(r_i,s_i)} \right\rangle = \sum_k T_k F_k \,. \tag{3.29}$$

The coefficients $F_k$ are still solutions of crossing symmetry, and we conjecture that they generate the space of solutions. In other words, we conjecture that all solutions of the crossing symmetry equations belong to the $O(n)$ CFT. This conjecture is natural, because the nontrivial input in the crossing symmetry equations is the spectrum of the model. It would be interesting to test this conjecture in simpler cases, such as minimal models: in this case, the conjecture says that we would not find more solutions by allowing channel fields to violate fusion rules, while still belonging to the spectrum.

The conjecture would be wrong if we were only considering two-channel crossing symmetry equations: we could then find solutions that have nothing to do with the $O(n)$ CFT, as is known to happen in the $Q$-state Potts model [31, 32]. But the three-channel equations (3.16) are more constraining. All our numerical results will support the conjecture.

From the conjecture, it follows that the number of invariants provides an upper bound on the number of solutions of crossing symmetry,

$$\boxed{\mathcal{N}_{\left\langle \prod_{i=1}^{4} V_{(r_i,s_i)} \right\rangle} \leq \mathcal{I}_{\left\langle \prod_{i=1}^{4} V^{\Lambda_{(r_i,s_i)}} \right\rangle}} \,. \tag{3.30}$$

We expect an inequality, but not necessarily an equality, because two solutions can coincide by a dynamical accident. In numerical results, we will indeed find many cases where the inequality is strict.

**Fusion rules**

In order to explicitly determine a particular solution $F_k$ of crossing symmetry, we should use the existence of fusion rules in our CFT,

$$V^{\lambda_1}_{(r_1,s_1)} \times V^{\lambda_2}_{(r_2,s_2)} \sim \delta_{r_1,r_2}\delta_{s_1-s_2 \in 2\mathbb{Z}}\delta_{[]\subset \lambda_1 \otimes \lambda_2} \sum_{s \in |s_1-s_2|+1+2\mathbb{N}} V^{D}_{\langle 1,s\rangle}$$

$$+ \sum_{r \in \frac{1}{2}\mathbb{N}^* \cap (\mathbb{Z}+r_1+r_2)} \sum_{s \in \frac{\mathbb{Z}}{r}} \sum_{\nu \subset \Lambda_{(r,s)} \cap (\lambda_1 \otimes \lambda_2)} V^{\nu}_{(r,s)} \,. \tag{3.31}$$

These fusion rules take into account the conformal fusion rules (3.17), plus the constraints of $O(n)$ symmetry, and the condition $v \subset \Lambda_{(r,s)}$ from the structure of the spectrum (2.20). In the case of the four-point function $\left\langle \prod_{i=1}^{4} V_{(\frac{1}{2},0)} \right\rangle$, the solution $F_{[]}^{(s)}$, which corresponds to the invariant tensor $T_{[]}^{(s)}$, may involve the $s$-channel Virasoro representation $\mathcal{W}_{(2,0)}$ since $[] \subset \Lambda_{(2,0)}$ (2.27c), but not the representations $\mathcal{W}_{(1,0)}$, $\mathcal{W}_{(1,1)}$, $\mathcal{W}_{(2,\frac{1}{2})}$ and $\mathcal{W}_{(2,1)}$. Removing these representations from the spectrum $\mathcal{S}^{(s)}$ in the crossing symmetry equations (3.16) reduces the dimension of the space of solutions from 3 to 1, singling out the desired solution.

This type of reasoning typically determines a solution $F_v^{(x)}$ (with $x \in \{s, t, u\}$) up to an overall normalization. To fix this normalization, we can use the relations between the three bases of solutions. For example, the bases of invariants $\left\{T_v^{(s)}\right\}_v$ and $\left\{T_v^{(t)}\right\}_v$ obey a linear relation of the type $T_v^{(s)} = \sum_\lambda M_{v\lambda} T_\lambda^{(t)}$, whose coefficients are rational functions of $n$ [35]. Therefore, the corresponding bases of solutions $\left\{F_v^{(s)}\right\}_v$ and $\left\{F_v^{(t)}\right\}_v$ must obey a similar linear relation, whose matrix is $M^{-1T}$. We expect that this linear relation fixes the relative normalizations of the solutions, which determines the four-point function (3.29) up to one overall constant coefficient. We will sketch this fixing of normalizations in an example in Section 4.1.

**OPE commutativity and parity of spins**

The fusion rule of two identical fields $V_{(r_1,s_1)}^{\lambda_1} \times V_{(r_1,s_1)}^{\lambda_1}$ obeys an extra constraint, because of OPE commutativity. Exchanging the two fields, the term of $V_{(r,s)}^v$ in the OPE picks a factor $(-1)^{rs}$ from the dependence on field positions, and a factor $\epsilon_{\lambda_1}(v) \in \{-1, 1\}$ from the Clebsch-Gordan coefficient. For example, since the map

$$\left(v_i \otimes v_j \mapsto v_i \otimes v_j - v_j \otimes v_i\right) \in \mathrm{Hom}([1] \otimes [1], [11]), \tag{3.32}$$

is antisymmetric under $i \leftrightarrow j$, we have $\epsilon_{[1]}([11]) = -1$. OPE commutativity then implies

$$V_{(r,s)}^v \in V_{(r_1,s_1)}^{\lambda_1} \times V_{(r_1,s_1)}^{\lambda_1} \implies (-1)^{rs} \epsilon_{\lambda_1}(v) = 1. \tag{3.33}$$

This means that a given representation $v$ can only be associated to fields with conformal spins that are either odd, or even.

But how do we determine $\epsilon_{\lambda_1}(v)$? This quantity may actually be ambiguous if $v$ appears several times in $\lambda_1 \otimes \lambda_1$. We did not find general results on this subject. In Section 4.3, we will find that the constraint (3.33) is obeyed in numerical bootstrap results. For $\lambda_1 \in \{[1], [11], [2]\}$, we found that $\epsilon_{\lambda_1}(v)$ actually does not depend on $\lambda_1$, and

$$\lambda_1 \in \{[1], [11], [2]\} \implies \begin{cases} \epsilon_{\lambda_1}([]) = \epsilon_{\lambda_1}([2]) = \epsilon_{\lambda_1}([4]) = \epsilon_{\lambda_1}([22]) = \epsilon_{\lambda_1}([1111]) = 1, \\ \epsilon_{\lambda_1}([11]) = \epsilon_{\lambda_1}([211]) = \epsilon_{\lambda_1}([31]) = -1. \end{cases}$$
$$\tag{3.34}$$

## 3.3 Numerical implementation

Our numerical treatment of crossing symmetry equations follows the method of [28], which is applicable when the spectrum of conformal dimensions is known exactly. The basic idea is to truncate the equations to a finite system, and to deduce the existence of exact solutions from the behaviour of the truncated system's solutions as the cutoff increases. In order to reduce the size of the system, we follow [31] and use interchiral blocks rather than conformal blocks. We further reduce computing time by drawing only one set of random positions rather than several.

**Truncated crossing symmetry equations**

We are interested in solving the crossing equation (3.16) for the structure constants at generic central charge. Since Virasoro conformal blocks have poles at rational values of the central charge, we choose values $c \in \mathbb{C} - \mathbb{R}$.

We group the fields in the spectrum into infinite families, whose relative structure constants are fixed by shift equations from degenerate fields. At the level of conformal blocks, this amounts to replacing conformal blocks with interchiral blocks. The number of families is still infinite, so we need to truncate the spectrum using a cutoff on the total conformal dimension

$$\Re(\Delta + \bar{\Delta}) \leq \Delta_{\text{max}} . \tag{3.35}$$

Given the structure of the spectrum (2.20), and the assumption (1.4) on the parameter $\beta^2$, this truncation leaves us with a finite number $N_{\text{unknowns}}$ of "interchiral primary" fields. The truncation can also be applied to descendant fields when computing a conformal block, reducing this computation to summing a finite series.

Having made the number of unknowns $N_{\text{unknowns}}$ finite, we can now afford to make the number of equations finite as well. In the three-channel bootstrap, there are in principle two equations for each value of the cross-ratio $z \in \mathbb{C}$ of the four fields' positions. Choosing $2N_{\text{cross-ratios}} = N_{\text{unknowns}} - 1$ would give us a unique solution up to an overall factor. But we still need to test whether this solves crossing symmetry at other values of $z$. In [28] this was done by looking at how the solution depends on the random draw of positions $\{z_k\}_{k=1,2,\dots,N_{\text{cross-ratios}}}$. However, it is computationally wasteful to generate the system again with completely different positions.

Instead, we propose to add only a few more positions than necessary, i.e. to choose $2N_{\text{cross-ratios}} = N_{\text{unknowns}} + O(1)$. We can then compare the solution of the first $N_{\text{unknowns}} - 1$ equations, with the solution of the last $N_{\text{unknowns}} - 1$ equations. The relative difference of these two solutions is called the **deviation**. We have a good determination of a non-vanishing structure constant if its deviation goes to zero as $\Delta_{\text{max}} \to \infty$. We have a solution of crossing symmetry if the deviation of any given non-vanishing structure constant goes to zero. In practice, a deviation that is much smaller than 1 usually indicates that the corresponding structure constant is nonzero, and that we know its value with a relative error of the order of the deviation.

Let us give more details on the structure of our linear system of crossing symmetry equations (3.16). After grouping the fields and truncating, the spectrum in each channel is made of finitely many interchiral primary fields, which we write as $V_1, V_2, V_3, \cdots$. The equations' unknowns are four-point structure constants $D_{V_k}^{(x)}$ with $x \in \{s, t, u\}$, and we build the vector of unknowns by stacking the unknowns from each channel:

$$\vec{d} = \begin{pmatrix} d^{(s)} \\ d^{(t)} \\ d^{(u)} \end{pmatrix} \quad \text{with} \quad d^{(x)} = \begin{bmatrix} D_{V_1}^{(x)} \\ D_{V_2}^{(x)} \\ D_{V_3}^{(x)} \\ \vdots \end{bmatrix} . \tag{3.36}$$

In total, the vector $\vec{d}$ has $N_{\text{unknowns}}$ components, split in some way between the three channels. From the values of the interchiral blocks, we then build three matrices $b^{(s)}, b^{(t)}, b^{(u)}$,

$$b^{(x)} = \begin{bmatrix} \mathcal{H}_{V_1}^{(x)}(z_1) & \mathcal{H}_{V_2}^{(x)}(z_1) & \mathcal{H}_{V_3}^{(x)}(z_1) & \dots \\ \mathcal{H}_{V_1}^{(x)}(z_2) & \mathcal{H}_{V_2}^{(x)}(z_2) & \mathcal{H}_{V_3}^{(x)}(z_2) & \dots \\ \vdots & \vdots & \vdots & \end{bmatrix} . \tag{3.37}$$

The number of lines of these matrices is $N_{\text{cross-ratios}}$. From these three matrices, we build the block matrix

$$B = \begin{pmatrix} b^{(s)} & -b^{(t)} & 0 \\ 0 & b^{(t)} & -b^{(u)} \end{pmatrix} , \tag{3.38}$$

and the crossing symmetry equations (3.16) now take the form

$$B\vec{d} = 0 . \tag{3.39}$$

The same equation could equivalently be written using the alternative block matrices

$$B = \begin{pmatrix} b^{(s)} & -b^{(t)} & 0 \\ b^{(s)} & 0 & -b^{(u)} \end{pmatrix} \quad \text{or} \quad B = \begin{pmatrix} 0 & b^{(t)} & -b^{(u)} \\ b^{(s)} & 0 & -b^{(u)} \end{pmatrix} . \tag{3.40}$$

**Singular values**

In principle, the number $\mathcal{N}_{\left\langle \prod_{i=1}^{4} V_{(r_i, s_i)} \right\rangle}$ of solutions of the crossing-symmetry equations is the number of vanishing singular values of the untruncated crossing matrix, i.e., of the analogue of $B$ before truncation. However, after truncating the system, no singular value is exactly zero. We should therefore count singular values that are very small and/or that go to zero as $\Delta_{\max} \to \infty$. This is not necessarily straightforward, because large matrices tend to have small singular values, even if they are not degenerate.

We do not have a mathematically well-founded criterion for determining which singular values indicate the existence of bootstrap solutions. However, our experience with numerical data suggests that it is possible to deduce the number of solutions from the singular values of the crossing matrix $B$ (3.38), provided the cutoff is large enough.

Let us demonstrate this in two examples, by plotting the few lowest singular values as functions of $\Delta_{\max}$. Here and in other numerical examples, we choose $\beta^{-1} = 0.8 + 0.1i$: a generic complex value of $\beta$, far enough from the singularities of the conformal blocks at $\beta^2 \in \mathbb{Q}$. In each case we draw in red the singular values that correspond to bootstrap solutions, as we see from the large and increasing gap that separates them from the other singular values. The first plot is for the simplest nontrivial four-point function $\left\langle V_{(\frac{1}{2},0)} V_{(\frac{1}{2},0)} V_{(\frac{1}{2},0)} V_{(\frac{1}{2},0)} \right\rangle$, where we have 3 solutions:

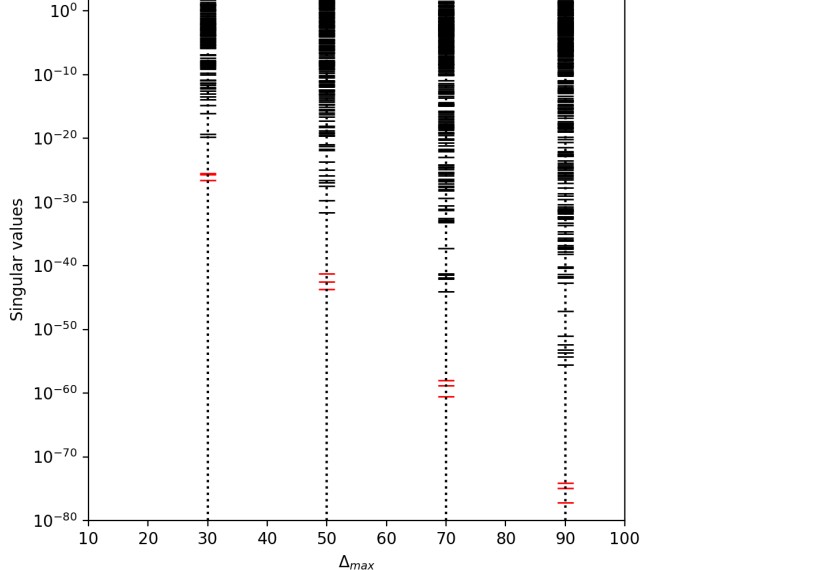

$$\tag{3.41}$$

In the case of the four-point function $\left\langle V_{(\frac{3}{2},0)}V_{(\frac{3}{2},0)}V_{(\frac{1}{2},0)}V_{(\frac{1}{2},0)}\right\rangle$, the separation between the 5 lowest singular values and the rest is not large for low values of $\Delta_{\max}$, but increases as $\Delta_{\max} \to \infty$:

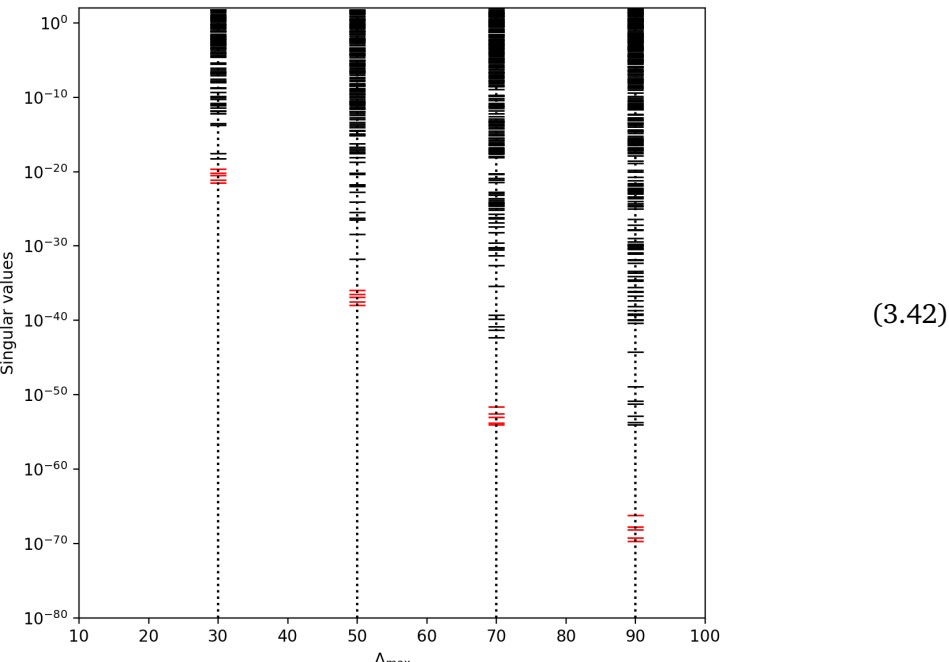

$$(3.42)$$

Counting solutions by examining singular values is therefore possible, but can take considerable amounts of resources because we need to compute the crossing matrix $B$ at high precision. For instance, the crossing matrix for Figure (3.41) at $\Delta_{\max} = 90$ has a precision of around 70 digits. The computation took a standard desktop computer two days. And the needed precision increases with the number of solutions. To evade this issue, we will now introduce another method for counting solutions, which requires less computing power—but more human craftsmanship.

**Method of excluding fields**

In our bootstrap method, it is easy to detect whether the solution of the crossing symmetry equations is unique: when this happens, the deviations of non-vanishing structure constants go to zero. We can therefore single out a particular solution by adding constraints until a unique solution is found. The number of independent solutions is the number of constraints. In practice, the constraints that we use consist in setting structure constants to zero, i.e., excluding fields from the spectrum. (We also set one structure constant to one as a normalization condition.)

This method is however more an art than a science, due to the nontrivial structure of the bootstrap solutions. It can indeed happen that a structure constant (or a linear combination of structure constants) vanishes in all solutions of a given system of crossing symmetry equations. When this happens, setting that structure constant to zero does not reduce the number of solutions, and could therefore lead to miscounting.

In practice, it is rather easy to count solutions in any given example by this method, but we do not know how to automate the process. Excluding fields has the added advantage of singling out specific solutions of crossing symmetry.

Let us illustrate this method in the case of $\left\langle V_{(\frac{1}{2},0)}^4 \right\rangle$ at $\Delta_{\max} = 40$ and $\beta^{-1} = 0.8 + 0.1i$. We will display the first few $s$-channel structure constants and their deviations, omitting the $t$- and $u$-channel structure constants for brevity. For brevity again, we only display real parts of structure constants, giving only one or two significant digits. We adopt the normalization condition $D_1^{(s)} = 1$, i.e., we normalize the structure constant of the identity field in the $s$-channel. We display the data before and after excluding the fields $V_{(1,0)}$ and $V_{(1,1)}$:

| | Before | | | After | |
|---|---|---|---|---|---|
| $(r,s)$ | $\Re D_{(r,s)}^{(s)}$ | Deviation | | $\Re D_{(r,s)}^{(s)}$ | Deviation |
| $\langle 1,1 \rangle$ | 1 | 0 | | 1 | 0 |
| $(1,0)$ | 1.24 | 0.16 | | $-$ | $-$ |
| $(1,1)$ | $-0.029$ | 0.15 | | $-$ | $-$ |
| $(2,0)$ | $-8.9 \times 10^{-4}$ | 0.14 | | $-1.5 \times 10^{-3}$ | $1.5 \times 10^{-19}$ |
| $(2,\pm\frac{1}{2})$ | $0.3 \times 10^{-3}$ | 0.15 | | $-1.1 \times 10^{-21}$ | 0.21 |
| $(2,1)$ | $-2 \times 10^{-3}$ | 0.16 | | $-2.9 \times 10^{-22}$ | 0.24 |
| $(3,0)$ | $2.8 \times 10^{-7}$ | 0.15 | | $1.3 \times 10^{-7}$ | $6.7 \times 10^{-11}$ |
| $(3,\pm\frac{1}{3})$ | $-8.0 \times 10^{-8}$ | 0.15 | | $-1.7 \times 10^{-18}$ | 2.6 |
| $(3,\pm\frac{2}{3})$ | $2.8 \times 10^{-7}$ | 0.15 | | $8.3 \times 10^{-8}$ | $7.4 \times 10^{-11}$ |

$$(3.43)$$

Before excluding two fields, all deviations are large: we have not singled out a solution. So we have to exclude some fields. We may choose the excluded fields arbitrarily: for definiteness we choose them by increasing values of the conformal dimension. The field with the lowest dimension after the identity field is $V_{(1,0)}$: excluding it however does not improve the deviations. If we moreover exclude the next field $V_{(1,1)}$, then the fields $V_{(2,0)}, V_{(3,0)}$ and $V_{(3,\pm\frac{2}{3})}$ have small deviations, while $V_{(2,\pm\frac{1}{2})}, V_{(2,1)}$ and $V_{(3,\pm\frac{1}{3})}$ have small values and large deviations. This signals the existence of a crossing symmetry solution with $D_{(2,\pm\frac{1}{2})}^{(s)} = D_{(2,1)}^{(s)} = D_{(3,\pm\frac{1}{3})}^{(s)} = 0$, whereas $D_{(2,0)}^{(s)}, D_{(3,0)}^{(s)}$ and $D_{(3,\pm\frac{2}{3})}^{(s)}$ have non-vanishing values that are well approximated by our numerical results, with relative errors of the order of their respective deviations. For example, we know $D_{(3,0)}^{(s)}$ with about 10 significant digits, i.e. with an absolute error $O(10^{-17})$. This solution is called $F_{[]}^{(s)}$.

**Precision of the results**

Finally, given a solution of crossing symmetry as a list of structure constants, we can compute the corresponding four-point function in all three channels. After all, the four-point function is the physical observable: in the $Q$-state Potts model, some four-point functions can be compared to results from Monte-Carlo simulations [15], and this should be doable in the $O(n)$ CFT too. Moreover, we can compare the results from the three channels, and directly check that crossing symmetry is obeyed at arbitrary values of the cross-ratio.

For example, here are the relative differences between the $s$, $t$ and $u$-channel calculations

of $F_{[]}^{(s)}$ at $\beta^{-1} = 0.8 + 0.1i$ and $z = 0.3 + 0.1i$, depending on $\Delta_{\max}$:

| $\Delta_{\max}$ | $s$ versus $t$ | $t$ versus $u$ |
|---|---|---|
| 30 | $2.7 \times 10^{-23}$ | $3.6 \times 10^{-22}$ |
| 50 | $2.7 \times 10^{-39}$ | $4.3 \times 10^{-37}$ |
| 70 | $1 \times 10^{-53}$ | $2.2 \times 10^{-51}$ |
| 90 | $1 \times 10^{-66}$ | $3.7 \times 10^{-64}$ |

$$(3.44)$$

The relative differences decrease exponentially with $\Delta_{\max}$, which confirms that the results are converging towards an exact solution.

# 4 Solutions of crossing symmetry equations

The numerical results in this section were obtained using publicly available Python code [33]. In particular, relatively low-precision results for all considered solutions are found in the notebook `On4pt.ipynb`.

Let us introduce notations for writing which fields appear in a given correlation function or fusion rule. These fields must belong to the spectrum $\mathcal{S}^{O(n)}$ (2.20). However, due to the conservation of $r$ modulo integers, it is convenient to introduce subspectra with values of $r$ that differ by integers. For any $\ell \in \frac{1}{2}\mathbb{N}^*$, we introduce

$$\mathcal{S}_\ell = \left\{ (r,s) \in (\mathbb{N} + \ell) \times (-1, 1] \Big| rs \in \mathbb{Z} \right\} . \tag{4.1}$$

Moreover, let $\mathcal{S}_0$ be $\mathcal{S}_1$ plus degenerate fields. While $\mathcal{S}^{O(n)}$ was initially defined as a vector space, we now identify it with a set of Kac indices for the corresponding indecomposable representations of the interchiral algebra, hence $s \in (-1, 1]$. Furthermore, the commutativity of OPEs of identical fields (3.33) suggests the further split $\mathcal{S}_1 = \mathcal{S}_{\text{even}} \sqcup \mathcal{S}_{\text{odd}}$ according to the parity of the conformal spin, with

$$\mathcal{S}_{\text{even}} = \{(r,s) \in \mathbb{N} \times (-1, 1] | rs \in 2\mathbb{Z}\} , \tag{4.2}$$

$$\mathcal{S}_{\text{odd}} = \{(r,s) \in \mathbb{N}^* \times (-1, 1] | rs \in 2\mathbb{Z} + 1\} . \tag{4.3}$$

It is also useful to list which indices are relevant to a given $O(n)$ representation,

$$\mathcal{S}^\lambda = \left\{ (r,s) \in \mathcal{S}^{O(n)} \Big| \lambda \subset \Lambda_{(r,s)} \right\} . \tag{4.4}$$

Based on our conjecture (2.26) for $\Lambda_{(r,s)}$, we further conjecture

$$\mathcal{S}^\lambda = \mathcal{S}_{\frac{1}{2}|\lambda|} - F_\lambda \quad \text{with} \quad F_\lambda \text{ finite} . \tag{4.5}$$

When examining the expressions (2.27) for $\Lambda_{(r,s)}$ with $r \le 3$, and with quite a few higher values of $r$, we indeed find that the multiplicity of any given irreducible representation $\lambda$ tends to increase with $r$. While the increase is not always monotonous, sharp decreases do not occur, and the multiplicity does not drop to zero after reaching high enough values. For example, the multiplicities $m_{(r,s)}^{[2]}$ of $\lambda = [2]$ in $\Lambda_{(r,s)}$ with $r \le 4$ are:

| $(r,s)$ | $(1,0)$ | $(1,1)$ | $(2,0)$ | $(2,\pm\frac{1}{2})$ | $(2,1)$ | $(3,0)$ | $(3,\pm\frac{1}{3})$ | $(3,\pm\frac{2}{3})$ | $(3,1)$ | $(4,0)$ | $(4,\pm\frac{1}{4})$ | $(4,\pm\frac{1}{2})$ | $(4,1)$ |
|---|---|---|---|---|---|---|---|---|---|---|---|---|---|
| $m_{(r,s)}^{[2]}$ | 1 | 0 | 1 | 0 | 1 | 4 | 2 | 4 | 2 | 23 | 18 | 23 | 23 |

Then we find $m_{(5,s)}^{[2]} \geq 174$ and it is hard to imagine $m_{(r,s)}^{[2]}$ reaching zero ever after. Therefore, we determine $F_\lambda$ by examining $\Lambda_{(r,s)}$ with the first few values of $r \geq \frac{1}{2}|\lambda|$, and we find:

$$\mathcal{S}^{[]} = \mathcal{S}_0 - \left\{(1,0),(1,1),(2,\pm\tfrac{1}{2}),(2,1),(3,\pm\tfrac{1}{3}),(3,1)\right\}, \tag{4.6a}$$

$$\mathcal{S}^{[1]} = \mathcal{S}_{\frac{1}{2}} - \left\{(\tfrac{3}{2},0),(\tfrac{3}{2},\pm\tfrac{2}{3})\right\}, \tag{4.6b}$$

$$\mathcal{S}^{[2]} = \mathcal{S}_1 - \left\{(1,1),(2,\pm\tfrac{1}{2})\right\}, \tag{4.6c}$$

$$\mathcal{S}^{[11]} = \mathcal{S}_1 - \left\{(1,0),(2,0),(2,1)\right\}, \tag{4.6d}$$

$$\mathcal{S}^{[3]} = \mathcal{S}_{\frac{3}{2}} - \left\{(\tfrac{3}{2},\pm\tfrac{2}{3})\right\}, \tag{4.6e}$$

$$\mathcal{S}^{[21]} = \mathcal{S}_{\frac{3}{2}} - \left\{(\tfrac{3}{2},0)\right\}, \tag{4.6f}$$

$$\mathcal{S}^{[111]} = \mathcal{S}_{\frac{3}{2}} - \left\{(\tfrac{3}{2},\pm\tfrac{2}{3})\right\}, \tag{4.6g}$$

$$\mathcal{S}^{[4]} = \mathcal{S}_2 - \left\{(2,\pm\tfrac{1}{2}),(2,1)\right\}, \tag{4.6h}$$

$$\mathcal{S}^{[31]} = \mathcal{S}_2 - \left\{(2,0)\right\}, \tag{4.6i}$$

$$\mathcal{S}^{[22]} = \mathcal{S}_2 - \left\{(2,\pm\tfrac{1}{2})\right\}, \tag{4.6j}$$

$$\mathcal{S}^{[211]} = \mathcal{S}_2 - \left\{(2,1)\right\}, \tag{4.6k}$$

$$\mathcal{S}^{[1111]} = \mathcal{S}_2 - \left\{(2,0),(2,\pm\tfrac{1}{2})\right\}. \tag{4.6l}$$

Notice the equality $\mathcal{S}^{[3]} = \mathcal{S}^{[111]}$.

## 4.1 The simplest four-point function $\left\langle V_{(\frac{1}{2},0)}^4 \right\rangle$

**Invariants and bases**

From the point of view of $O(n)$ representations, and with $O(n)$ vector indices explicit, our four-point function reads $\left\langle V_{i_1}^{[1]} V_{i_2}^{[1]} V_{i_3}^{[1]} V_{i_4}^{[1]} \right\rangle$. The $s$-channel invariants read

$$T_{[]}^{(s)} = \delta_{i_1 i_2}\delta_{i_3 i_4}, \tag{4.7a}$$

$$T_{[11]}^{(s)} = \delta_{i_1 i_4}\delta_{i_2 i_3} - \delta_{i_1 i_3}\delta_{i_2 i_4}, \tag{4.7b}$$

$$T_{[2]}^{(s)} = \delta_{i_1 i_3}\delta_{i_2 i_4} + \delta_{i_1 i_4}\delta_{i_2 i_3} - \frac{2}{n}\delta_{i_1 i_2}\delta_{i_3 i_4}. \tag{4.7c}$$

The decomposition (3.29) of our four-point function over this basis predicts three solutions of crossing symmetry $F_{[]}^{(s)}, F_{[2]}^{(s)}, F_{[11]}^{(s)}$. Consider the other basis of invariants,



$$T_{[]}^{(s)} = \delta_{i_1 i_2}\delta_{i_3 i_4} \qquad T_{[]}^{(t)} = \delta_{i_2 i_3}\delta_{i_1 i_4} \qquad T_{[]}^{(u)} = \delta_{i_1 i_3}\delta_{i_2 i_4} \tag{4.8}$$

and let $F_0^{(s)}, F_0^{(t)}, F_0^{(u)}$ be the corresponding solutions of crossing symmetry. From the linear relations between the invariants, we deduce

$$F_0^{(s)} = F_{[]}^{(s)} - \frac{2}{n}F_{[2]}^{(s)}, \quad F_0^{(t)} = F_{[2]}^{(s)} + F_{[11]}^{(s)}, \quad F_0^{(u)} = F_{[2]}^{(s)} - F_{[11]}^{(s)}. \tag{4.9}$$

From the definition (3.29), our four-point function is

$$\left\langle V_{i_1}^{[1]} V_{i_2}^{[1]} V_{i_3}^{[1]} V_{i_4}^{[1]} \right\rangle = \sum_{x \in \{s,t,u\}} T_{[]}^{(x)} F_0^{(x)} . \tag{4.10}$$

The four-point function must be covariant under permutations of the four fields, and therefore also under permutations of the three channels. This implies that the three solutions $F_0^{(x)}$ are related by permutations, just like the three invariants $T_{[]}^{(x)}$. In particular, $D_{(r,s)}^{(t)}(F_0^{(t)}) = D_{(r,s)}^{(s)}(F_0^{(s)})$. Using Eq. (4.9), this allows us to fix the relative normalizations of the three solutions $F_{[]}^{(s)}, F_{[2]}^{(s)}, F_{[11]}^{(s)}$. For example, we know that $D_{\text{odd spin}}^{(s)}(F_0^{(s)}) = 0$, which implies $D_{\text{odd spin}}^{(t)}(F_{[2]}^{(s)} + F_{[11]}^{(s)}) = 0$, and therefore fixes the relative normalizations of $F_{[2]}^{(s)}$ and $F_{[11]}^{(s)}$. This determines the four-point function $\left\langle V_{i_1}^{[1]} V_{i_2}^{[1]} V_{i_3}^{[1]} V_{i_4}^{[1]} \right\rangle$ up to an overall constant prefactor.

**Numerical results**

From the rules (3.17) and (3.33), the representations that may appear in the $s$-channel decompositions of the solutions $F_{[]}^{(s)}, F_{[2]}^{(s)}, F_{[11]}^{(s)}$ are

$$\mathcal{S}^{(s)}\left(F_{[]}^{(s)}\right) = \mathcal{S}_{\text{even}}^{[]}, \quad \mathcal{S}^{(s)}\left(F_{[2]}^{(s)}\right) = \mathcal{S}_{\text{even}}^{[2]}, \quad \mathcal{S}^{(s)}\left(F_{[11]}^{(s)}\right) = \mathcal{S}_{\text{odd}}^{[11]} . \tag{4.11}$$

And indeed, for each one of these three $s$-channel spectra, we find a unique solution of crossing symmetry. We are able to single out these solutions in the 3-dimensional space of solutions because each spectrum involves setting two of the three structure constants $D_1^{(s)}, D_{(1,0)}^{(s)}, D_{(1,1)}^{(s)}$ to zero. In the $t$- and $u$-channels, these solutions have the largest possible spectra,

$$\mathcal{S}^{(t,u)}\left(F_{\lambda}^{(s)}\right) = \mathcal{S}_0 , \tag{4.12}$$

i.e. anything that is allowed by the conservation of $r$ mod $\mathbb{Z}$. Let us display numerical data that underlie these results. We list all $s$-channel fields with small deviations $< 0.1$, together with the first field with a large deviation for comparison. For each field with a small deviation, we display the real part of the four-point structure constant with as many digits as the deviation suggests are significant. For example, in the case of $F_{[]}^{(s)}$, although $D_{(3,0)}^{(s)} = O(10^{-15})$ is rather small, we do know its value with about 15 significant digits, i.e., the absolute error is $O(10^{-30})$.

$$F_{[]}^{(s)} \text{ at } \Delta_{\max} = 40 \text{ and } \beta^{-1} = 0.8 + 0.1i,$$

| $(r,s)$ | $\Re D_{(r,s)}^{(s)}$ | Deviation |
|---|---:|---:|
| $\langle 1,1 \rangle$ | 1 | 0 |
| $(2,0)$ | $-1.515508647813802768 \times 10^{-3}$ | $2.2 \times 10^{-19}$ |
| $(3,0)$ | $1.39468476197762 \times 10^{-15}$ | $6.6 \times 10^{-15}$ |
| $(3, \pm \frac{2}{3})$ | $8.3751227046841 \times 10^{-8}$ | $1.2 \times 10^{-14}$ |
| $(4,0)$ | $3.7 \times 10^{-13}$ | $0.87$ |

$$(4.13)$$

$$F_{[11]}^{(s)} \text{ at } \Delta_{\max} = 40 \text{ and } \beta^{-1} = 0.8 + 0.1i \,,$$

| $(r,s)$ | $\Re D_{(r,s)}^{(s)}$ | Deviation |
|---|---|---|
| $(1,1)$ | $1$ | $0$ |
| $(2,\pm\frac{1}{2})$ | $-1.136421079784788769 \times 10^{-3}$ | $4.7 \times 10^{-20}$ |
| $(3,\pm\frac{1}{3})$ | $4.60859597460550 \times 10^{-7}$ | $4.1 \times 10^{-15}$ |
| $(3,1)$ | $2.43369637600654 \times 10^{-7}$ | $3.1 \times 10^{-15}$ |
| $(4,\pm\frac{1}{4})$ | $5.8 \times 10^{-13}$ | $0.48$ |

(4.14)

$$F_{[2]}^{(s)} \text{ at } \Delta_{\max} = 40 \text{ and } \beta^{-1} = 0.8 + 0.1i \,,$$

| $(r,s)$ | $\Re D_{(r,s)}^{(s)}$ | Deviation |
|---|---|---|
| $(1,0)$ | $1$ | $0$ |
| $(2,0)$ | $-6.249142617756265636 \times 10^{-3}$ | $2.1 \times 10^{-19}$ |
| $(2,1)$ | $-1.4658809155406988148 \times 10^{-3}$ | $7.9 \times 10^{-20}$ |
| $(3,0)$ | $2.06056149998946 \times 10^{-7}$ | $7.1 \times 10^{-15}$ |
| $(3,\pm\frac{2}{3})$ | $2.15149154275906 \times 10^{-8}$ | $3.9 \times 10^{-15}$ |
| $(4,0)$ | $6.1 \times 10^{-13}$ | $0.57$ |

(4.15)

**Fusion rule interpretation**

From the solutions of crossing symmetry in the $s$-channel basis, we deduce the fusion rule

$$\boxed{V_{(\frac{1}{2},0)}^{[1]} \times V_{(\frac{1}{2},0)}^{[1]} \sim \sum_{k \in \mathcal{S}_{\text{even}}^{[]}} V_k^{[]} + \sum_{k \in \mathcal{S}_{\text{even}}^{[2]}} V_k^{[2]} + \sum_{k \in \mathcal{S}_{\text{odd}}^{[11]}} V_k^{[11]}} \,. \tag{4.16}$$

In this case, the fusion rules coincide with what we would expect from the spectrum, after splitting it according to odd or even spins. In other words, all fields that are allowed by symmetry to appear, do in fact appear, i.e. they come with nonzero structure constants in the solutions $F_\lambda^{(s)}$. For instance, from (4.13), the field $V_{(2,1)}$ vanishes in the singlet-channel because it cannot be decomposed on to the singlet. In other fusion rules, we will however find examples of fields that could appear, but do not.

This provides a test of the conjectured spectrum. In particular, the conjecture predicts that the field $V_{(2,1)}^{[]}$ does not exist, see the expression (4.6a) for $\mathcal{S}^{[]}$. Finding such a field would have killed the conjecture. The power of this test is limited because $\mathcal{S}^{[]}$ contains all possible pairs $(r,s)$ with $r \in \mathbb{N}_{\geq 3}$.

## 4.2 Four-point functions and fusion rules of $V_{(\frac{1}{2},0)}$, $V_{(1,0)}$ and $V_{(1,1)}$

The simplest non-degenerate fields in the $O(n)$ CFT are $V_{(\frac{1}{2},0)}$, $V_{(1,0)}$ and $V_{(1,1)}$. Each one of these fields correspond to a single irreducible representation of $O(n)$. The fields should therefore be written as $V_{(\frac{1}{2},0)}^{[1]}$, $V_{(1,0)}^{[2]}$ and $V_{(1,1)}^{[11]}$, but we can omit the $O(n)$ representation labels without introducing ambiguities.

**The four-point functions $\left\langle V^2_{(1,0)} V^2_{(\frac{1}{2},0)} \right\rangle$ and $\left\langle V^2_{(1,1)} V^2_{(\frac{1}{2},0)} \right\rangle$**

For each one of these four-point functions, there exist three independent $O(n)$ invariants, and we numerically find three independent solutions of crossing symmetry. Let us write the $s$- and $t$-channel bases of solutions:

| 4-point function | $s$-channel solutions | $t$-channel solutions |
|---|---|---|
| $\left\langle V^2_{(1,0)} V^2_{(\frac{1}{2},0)} \right\rangle$ | $F^{(s)}_{[\,]}, F^{(s)}_{[2]}, F^{(s)}_{[11]}$ | $F^{(t)}_{[1]}, F^{(t)}_{[21]}, F^{(t)}_{[3]}$ |
| $\left\langle V^2_{(1,1)} V^2_{(\frac{1}{2},0)} \right\rangle$ | $G^{(s)}_{[\,]}, G^{(s)}_{[2]}, G^{(s)}_{[11]}$ | $G^{(t)}_{[1]}, G^{(t)}_{[21]}, G^{(t)}_{[111]}$ |

(4.17)

In the $s$-channel, each solution can be singled out by requiring the vanishing of two structure constants among $D^{(s)}_1$, $D^{(s)}_{(1,0)}$, $D^{(s)}_{(1,1)}$, see Eqs. (4.6a)-(4.6d). The situation is similar in the $t$-channel with $D^{(t)}_{(\frac{1}{2},0)}$, $D^{(t)}_{(\frac{3}{2},0)}$ and $D^{(t)}_{(\frac{3}{2},\frac{2}{3})}$.

We can therefore easily determine each solution numerically, and we find a nontrivial pattern of vanishing structure constants: all solutions obey $D^{(s)}_{(r,1)} = 0$ for $r \in \mathbb{N} + 2$. (Numerically, we are however limited to $r \leq 8$.) Let us display some of the structure constants in the singlet solution $F^{(s)}_{[\,]}$, computed at $\beta^{-1} = 0.8 + 0.1i$, to support our argument.

$$\Delta_{\max} = 40 \qquad\qquad\qquad \Delta_{\max} = 120$$

| $(r,s)$ | $\Re D^{(s)}_{(r,s)}$ | Deviation | $\Re D^{(s)}_{(r,s)}$ | Deviation |
|---|---|---|---|---|
| $\langle 1,1 \rangle$ | $1$ | $0$ | $1$ | $0$ |
| $(2,0)$ | $1.3371893\ldots \times 10^{-3}$ | $4.5 \times 10^{-25}$ | $1.3371893\ldots \times 10^{-3}$ | $6.6 \times 10^{-91}$ |
| $(2,1)$ | $3.0 \times 10^{-28}$ | $1.4$ | $7.5 \times 10^{-93}$ | $0.91$ |
| $(4,0)$ | $-2.7870591\ldots \times 10^{-13}$ | $3.0 \times 10^{-13}$ | $-2.7870591\ldots \times 10^{-13}$ | $2.2 \times 10^{-78}$ |
| $(4,1)$ | $-4.9 \times 10^{-27}$ | $0.92$ | $-5.0 \times 10^{-92}$ | $4.0$ |
| $(6,0)$ | $-6.5 \times 10^{-22}$ | $1.8$ | $-7.9541852\ldots \times 10^{-31}$ | $2.5 \times 10^{-57}$ |
| $(6,1)$ | $7.2 \times 10^{-24}$ | $1.9$ | $2.5 \times 10^{-87}$ | $0.78$ |
| $(8,0)$ | — | — | $2.864293\ldots \times 10^{-55}$ | $1.7 \times 10^{-24}$ |
| $(8,1)$ | — | — | $6.1 \times 10^{-80}$ | $1.8$ |

(4.18)

Moreover, the solutions $G_{[11]}^{(s)}$ and $G_{[1]}^{(t)}$ obey $D_{(\frac{5}{2},0)}^{(t,u)} = 0$. These results may be written as

| Solutions | $s$-channel | $t$-channel | $u$-channel |
|---|---|---|---|
| $F_{[]}^{(s)}$ | $\mathcal{S}_{\text{even}}^{[]} - (2\mathbb{N}+2, 1)$ | $\mathcal{S}_{\frac{1}{2}}$ | $\mathcal{S}_{\frac{1}{2}}$ |
| $F_{[2]}^{(s)}$ | $\mathcal{S}_{\text{even}}^{[2]} - (2\mathbb{N}+2, 1)$ | $\mathcal{S}_{\frac{1}{2}}$ | $\mathcal{S}_{\frac{1}{2}}$ |
| $F_{[11]}^{(s)}$ | $\mathcal{S}_{\text{odd}}^{[11]} - (2\mathbb{N}+3, 1)$ | $\mathcal{S}_{\frac{1}{2}}$ | $\mathcal{S}_{\frac{1}{2}}$ |
| $F_{\lambda}^{(t)}$ | $\mathcal{S}_0 - (\mathbb{N}+2, 1)$ | $\mathcal{S}^{\lambda}$ | $\mathcal{S}_{\frac{1}{2}}$ |
| $G_{[]}^{(s)}$ | $\mathcal{S}_{\text{even}}^{[]} - (2\mathbb{N}+2, 1)$ | $\mathcal{S}_{\frac{1}{2}}$ | $\mathcal{S}_{\frac{1}{2}}$ |
| $G_{[2]}^{(s)}$ | $\mathcal{S}_{\text{even}}^{[2]} - (2\mathbb{N}+2, 1)$ | $\mathcal{S}_{\frac{1}{2}}$ | $\mathcal{S}_{\frac{1}{2}}$ |
| $G_{[11]}^{(s)}$ | $\mathcal{S}_{\text{odd}}^{[11]} - (2\mathbb{N}+3, 1)$ | $\mathcal{S}_{\frac{1}{2}} - \{(\frac{5}{2},0)\}$ | $\mathcal{S}_{\frac{1}{2}} - \{(\frac{5}{2},0)\}$ |
| $G_{[1]}^{(t)}$ | $\mathcal{S}_0 - (\mathbb{N}+2, 1)$ | $\mathcal{S}^{[1]} - \{(\frac{5}{2},0)\}$ | $\mathcal{S}_{\frac{1}{2}} - \{(\frac{5}{2},0)\}$ |
| $G_{[21]}^{(t)}, G_{[111]}^{(t)}$ | $\mathcal{S}_0 - (\mathbb{N}+2, 1)$ | $\mathcal{S}^{\lambda}$ | $\mathcal{S}_{\frac{1}{2}}$ |

$$(4.19)$$

We deduce the fusion rules

$$V_{(1,0)}^{[2]} \times V_{(\frac{1}{2},0)}^{[1]} \sim \sum_{k \in \mathcal{S}^{[1]}} V_k^{[1]} + \sum_{k \in \mathcal{S}^{[3]}} V_k^{[3]} + \sum_{k \in \mathcal{S}^{[21]}} V_k^{[21]},\qquad (4.20)$$

$$V_{(1,1)}^{[11]} \times V_{(\frac{1}{2},0)}^{[1]} \sim \sum_{k \in \mathcal{S}^{[1]} - \{(\frac{5}{2},0)\}} V_k^{[1]} + \sum_{k \in \mathcal{S}^{[111]}} V_k^{[111]} + \sum_{k \in \mathcal{S}^{[21]}} V_k^{[21]}.\qquad (4.21)$$

These are the only fusion rules that we can deduce from $\left\langle V_{(1,0)}^2 V_{(\frac{1}{2},0)}^2 \right\rangle$ and $\left\langle V_{(1,1)}^2 V_{(\frac{1}{2},0)}^2 \right\rangle$. Our four-point function are indeed of the type $\langle V_1 V_1 V_2 V_2 \rangle$, which gives us access to $V_1 \times V_2$ and to $(V_1 \times V_1) \cap (V_2 \times V_2)$, from which we however cannot deduce $V_1 \times V_1$ or $V_2 \times V_2$.

**The four-point function $\left\langle V_{(1,0)}^2 V_{(1,1)}^2 \right\rangle$**

There exist four independent $O(n)$ invariants, and we numerically find four independent solutions of crossing symmetry. According to the tensor products (2.13f)-(2.13h) of the representations $[2]$ and $[11]$, the $s$- and $t$-channel bases of solutions are

| 4-point function | $s$-channel solutions | $t$-channel solutions |
|---|---|---|
| $\left\langle V_{(1,0)}^{[2]} V_{(1,0)}^{[2]} V_{(1,1)}^{[11]} V_{(1,1)}^{[11]} \right\rangle$ | $F_{[]}^{(s)}, F_{[2]}^{(s)}, F_{[11]}^{(s)}, F_{[22]}^{(s)}$ | $F_{[2]}^{(t)}, F_{[11]}^{(t)}, F_{[31]}^{(t)}, F_{[211]}^{(t)}$ |

$$(4.22)$$

It is easy to single out the solutions $F_{[11]}^{(t)}$, $F_{[31]}^{(t)}$ and $F_{[211]}^{(t)}$ by requiring the vanishing of three structure constants in each case among $\left\{ D_{(1,0)}^{(t)}, D_{(1,1)}^{(t)}, D_{(2,0)}^{(t)}, D_{(2,1)}^{(t)} \right\}$, see Eqs. (4.6d), (4.6i), (4.6k). However, this does not work for $F_{[2]}^{(t)}$: while $\mathcal{S}^{[2]}$ (4.6c) implies that $F_{[2]}^{(t)}$ obeys the three equations $D_{(1,1)}^{(t)} = D_{(2,\frac{1}{2})}^{(t)} = D_{(2,-\frac{1}{2})}^{(t)} = 0$, these equations are not all independent, because all solutions happen to obey $D_{(2,\frac{1}{2})}^{(t)} = D_{(2,-\frac{1}{2})}^{(t)}$.

To determine $F^{(t)}_{[2]}$, we have to think about the fusion rule $V_{(1,0)} \times V_{(1,1)}$, and to remember that this fusion rule also appears in the four-point function $\left\langle V_{(1,0)} V_{(1,1)} V^2_{(\frac{1}{2},0)} \right\rangle$. The numerical study of that four-point function shows that $V^{[2]}_{(2,0)}$ does not appear in the $s$-channel. However, $V^{[2]}_{(2,0)}$ does appear in the fusion rule $V_{(\frac{1}{2},0)} \times V_{(\frac{1}{2},0)}$ (4.16). Since there is a unique field of this type, it must be absent from the fusion rule $V_{(1,0)} \times V_{(1,1)}$, and therefore also from $F^{(t)}_{[2]}$. This provides the third constraint that we need for singling out the solution $F^{(t)}_{[2]}$. From the solutions $F^{(t)}_\lambda$, we then deduce the fusion rule

$$V^{[2]}_{(1,0)} \times V^{[11]}_{(1,1)} \sim \sum_{k \in \mathcal{S}^{[2]} - \{(2,0),(3,0)\}} V^{[2]}_k + \sum_{k \in \mathcal{S}^{[11]}} V^{[11]}_k + \sum_{k \in \mathcal{S}^{[31]}} V^{[31]}_k + \sum_{k \in \mathcal{S}^{[211]}} V^{[211]}_k . \tag{4.23}$$

**The fusion rules $V_{(1,0)} \times V_{(1,0)}$ and $V_{(1,1)} \times V_{(1,1)}$**

To determine these fusion rules, it would be enough to find the crossing symmetry solutions associated to the the four-point functions $\left\langle V^4_{(1,0)} \right\rangle$ and $\left\langle V^4_{(1,1)} \right\rangle$. In these cases, there exist six independent $O(n)$ invariants, and we numerically find six independent solutions of crossing symmetry. Let us write the $s$-channel bases of solutions:

| 4-point function | $s$-channel solutions |
|:---:|:---:|
| $\left\langle V^4_{(1,0)} \right\rangle$ | $F^{(s)}_{[]}, F^{(s)}_{[2]}, F^{(s)}_{[11]}, F^{(s)}_{[22]}, F^{(s)}_{[31]}, F^{(s)}_{[4]}$ |
| $\left\langle V^4_{(1,1)} \right\rangle$ | $G^{(s)}_{[]}, G^{(s)}_{[2]}, G^{(s)}_{[11]}, G^{(s)}_{[22]}, G^{(s)}_{[211]}, G^{(s)}_{[1111]}$ |

(4.24)

We can single out the solution $F^{(s)}_{[4]}$ by using the five independent constraints $D^{(s)}_1 = D^{(s)}_{(1,0)} = D^{(s)}_{(1,1)} = D^{(s)}_{(2,\frac{1}{2})} = D^{(s)}_{(2,1)} = 0$, see $\mathcal{S}^{[4]}$ (4.6h). In the case of the solution $F^{(s)}_{[31]}$, we have only four constraints, see $\mathcal{S}^{[31]}$ (4.6i), but we can add $D^{(s)}_{(2,1)} = 0$ thanks to the OPE commutativity rule (3.33) with $\epsilon_{[2]}([31]) = -1$. We are however unable to single out the remaining four solutions. To complete the determination of $V_{(1,0)} \times V_{(1,0)}$, we need to consider other correlation functions where this fusion rule appears.

Let us start with $\left\langle V^2_{(1,0)} V^2_{(\frac{1}{2},0)} \right\rangle$. We cannot deduce much from the nontrivial structures of the $s$-channel solutions (4.19): for example, since the field $V^{[2]}_{(4,1)}$ has nontrivial multiplicity in the spectrum, it could be absent from the $s$-channel solutions while being present in both fusion rules $V_{(1,0)} \times V_{(1,0)}$ and $V_{(\frac{1}{2},0)} \times V_{(\frac{1}{2},0)}$. The exception is $V^{[2]}_{(2,1)}$, which has multiplicity one. Since it appears in $V_{(\frac{1}{2},0)} \times V_{(\frac{1}{2},0)}$ but not in the $s$-channel of $\left\langle V^2_{(1,0)} V^2_{(\frac{1}{2},0)} \right\rangle$, it must be absent from $V_{(1,0)} \times V_{(1,0)}$.

This is just what we need for determining the $s$-channel solutions of $\left\langle V^2_{(1,0)} V^2_{(1,1)} \right\rangle$ (4.22). The solution that corresponds to $[2]$ is a priori problematic: imposing $D^{(s)}_{(1,1)} = 0$ brings us to the three-dimensional subspace of even-spin solutions, where $D^{(s)}_1 = 0$ is not enough for singling out our solution. The problem is solved by imposing $D^{(s)}_{(2,1)} = 0$. We then find that any field that is authorized by $O(n)$ symmetry and OPE commutativity does appear in the $s$-channel

of $\left\langle V_{(1,0)}^2 V_{(1,1)}^2 \right\rangle$, except for $V_{(2,1)}^{[2]}$. All these fields must therefore appear in the fusion rule

$$V_{(1,0)}^{[2]} \times V_{(1,0)}^{[2]} \sim \sum_{k\in\mathcal{S}_{\text{even}}^{[\,]}} V_k^{[\,]} + \sum_{k\in\mathcal{S}_{\text{even}}^{[2]}-\{(2,1)\}} V_k^{[2]} + \sum_{k\in\mathcal{S}_{\text{odd}}^{[11]}} V_k^{[11]} + \sum_{k\in\mathcal{S}_{\text{even}}^{[4]}} V_k^{[4]} + \sum_{k\in\mathcal{S}_{\text{odd}}^{[31]}} V_k^{[31]} + \sum_{k\in\mathcal{S}_{\text{even}}^{[22]}} V_k^{[22]} \,. \quad (4.25)$$

A similar reasoning allows us to determine

$$V_{(1,1)}^{[11]} \times V_{(1,1)}^{[11]} \sim \sum_{k\in\mathcal{S}_{\text{even}}^{[\,]}} V_k^{[\,]} + \sum_{k\in\mathcal{S}_{\text{even}}^{[2]}-\{(2,1)\}} V_k^{[2]} + \sum_{k\in\mathcal{S}_{\text{odd}}^{[11]}} V_k^{[11]} + \sum_{k\in\mathcal{S}_{\text{even}}^{[1111]}} V_k^{[1111]} + \sum_{k\in\mathcal{S}_{\text{odd}}^{[211]}} V_k^{[211]} + \sum_{k\in\mathcal{S}_{\text{even}}^{[22]}} V_k^{[22]} \,. \quad (4.26)$$

### 4.3 More examples

In the examples that we studied so far, the number of bootstrap solutions always matched the number of four-point invariants, i.e., the inequality (3.30) was saturated. We will now start a more systematic scan of four-point functions, and find examples where the inequality is strict.

**The 30 simplest four-point functions**

We organize correlators by increasing level $L = \sum_{i=1}^4 r_i$. There are 30 inequivalent four-point functions with $L = 2, 3, 4$. By inequivalent we mean not related by permutations, or by flipping the signs of all second indices $s_i$.

In each case, we indicate the number $\mathcal{N}$ of solutions of crossing symmetry, and the number $\mathcal{I}$ of $O(n)$ invariant tensors. We find that the inequality $\mathcal{N} \leq \mathcal{I}$ (3.30) is saturated in 21 out of 30 cases.

| Level-2 correlators | Bootstrap $\mathcal{N}$ | Invariants $\mathcal{I}$ |
|:---:|:---:|:---:|
| $\left(\frac{1}{2},0\right)^4$ | 3 | 3 |

| Level-3 correlators | Bootstrap $\mathcal{N}$ | Invariants $\mathcal{I}$ |
|:---:|:---:|:---:|
| $(1,0)^2\left(\frac{1}{2},0\right)^2$ | 3 | 3 |
| $(1,1)^2\left(\frac{1}{2},0\right)^2$ | 3 | 3 |
| $(1,0)(1,1)\left(\frac{1}{2},0\right)^2$ | 2 | 2 |
| $\left(\frac{3}{2},0\right)\left(\frac{1}{2},0\right)^3$ | 2 | 2 |
| $\left(\frac{3}{2},\frac{2}{3}\right)\left(\frac{1}{2},0\right)^3$ | 2 | 2 |

| Level-4 correlators | Bootstrap $\mathcal{N}$ | Invariants $\mathcal{I}$ |
|---|---|---|
| $(1,0)^4$ | 6 | 6 |
| $(1,0)^3(1,1)$ | 3 | 3 |
| $(1,0)^2(1,1)^2$ | 4 | 4 |
| $(1,0)(1,1)^3$ | 3 | 3 |
| $(1,1)^4$ | 6 | 6 |
| $\left(\frac{3}{2},0\right)(1,0)^2\left(\frac{1}{2},0\right)$ | 4 | 4 |
| $\left(\frac{3}{2},0\right)(1,0)(1,1)\left(\frac{1}{2},0\right)$ | 4 | 4 |
| $\left(\frac{3}{2},0\right)(1,1)^2\left(\frac{1}{2},0\right)$ | 4 | 4 |
| $\left(\frac{3}{2},\frac{2}{3}\right)(1,0)^2\left(\frac{1}{2},0\right)$ | 4 | 4 |
| $\left(\frac{3}{2},\frac{2}{3}\right)(1,0)(1,1)\left(\frac{1}{2},0\right)$ | 4 | 4 |
| $\left(\frac{3}{2},\frac{2}{3}\right)(1,1)^2\left(\frac{1}{2},0\right)$ | 4 | 4 |
| $\left(\frac{3}{2},0\right)^2\left(\frac{1}{2},0\right)^2$ | 5 | 6 |
| $\left(\frac{3}{2},0\right)\left(\frac{3}{2},\frac{2}{3}\right)\left(\frac{1}{2},0\right)^2$ | 4 | 4 |
| $\left(\frac{3}{2},\frac{2}{3}\right)^2\left(\frac{1}{2},0\right)^2$ | 5 | 5 |
| $\left(\frac{3}{2},\frac{2}{3}\right)\left(\frac{3}{2},-\frac{2}{3}\right)\left(\frac{1}{2},0\right)^2$ | 4 | 5 |
| $(2,0)(1,0)\left(\frac{1}{2},0\right)^2$ | 5 | 7 |
| $(2,0)(1,1)\left(\frac{1}{2},0\right)^2$ | 5 | 6 |
| $\left(2,\frac{1}{2}\right)(1,0)\left(\frac{1}{2},0\right)^2$ | 5 | 5 |
| $\left(2,\frac{1}{2}\right)(1,1)\left(\frac{1}{2},0\right)^2$ | 5 | 6 |
| $(2,1)(1,0)\left(\frac{1}{2},0\right)^2$ | 5 | 6 |
| $(2,1)(1,1)\left(\frac{1}{2},0\right)^2$ | 5 | 5 |
| $\left(\frac{5}{2},0\right)\left(\frac{1}{2},0\right)^3$ | 6 | 9 |
| $\left(\frac{5}{2},\frac{2}{5}\right)\left(\frac{1}{2},0\right)^3$ | 6 | 9 |
| $\left(\frac{5}{2},\frac{4}{5}\right)\left(\frac{1}{2},0\right)^3$ | 6 | 9 |

**Reducible versus irreducible representations:** $\left\langle V_{\left(\frac{3}{2},0\right)}V_{\left(\frac{1}{2},0\right)}^3\right\rangle$ **versus** $\left\langle V_{\left(\frac{3}{2},\frac{2}{3}\right)}V_{\left(\frac{1}{2},0\right)}^3\right\rangle$

The structures of $\Lambda_{\left(\frac{3}{2},0\right)}$ (2.27a) and $\Lambda_{\left(\frac{3}{2},\frac{2}{3}\right)}$ (2.27b) lead to the following predictions for the representations that may appear in the corresponding four-point functions with three $V_{\left(\frac{1}{2},0\right)}$ fields:

| 4-point function | $s$-channel | $t$-channel | $u$-channel |
|---|---|---|---|
| $\left\langle V_{\left(\frac{3}{2},0\right)}^{[3]} V_{\left(\frac{1}{2},0\right)}^{[1]} V_{\left(\frac{1}{2},0\right)}^{[1]} V_{\left(\frac{1}{2},0\right)}^{[1]}\right\rangle$ | $[2]$ | $[2]$ | $[2]$ |
| $\left\langle V_{\left(\frac{3}{2},0\right)}^{[111]} V_{\left(\frac{1}{2},0\right)}^{[1]} V_{\left(\frac{1}{2},0\right)}^{[1]} V_{\left(\frac{1}{2},0\right)}^{[1]}\right\rangle$ | $[11]$ | $[11]$ | $[11]$ |
| $\left\langle V_{\left(\frac{3}{2},\frac{2}{3}\right)}^{[21]} V_{\left(\frac{1}{2},0\right)}^{[1]} V_{\left(\frac{1}{2},0\right)}^{[1]} V_{\left(\frac{1}{2},0\right)}^{[1]}\right\rangle$ | $[2]+[11]$ | $[2]+[11]$ | $[2]+[11]$ |

(4.27)

And indeed, we numerically find that the bootstrap equations for $\left\langle V_{(\frac{3}{2},0)} V_{(\frac{1}{2},0)}^3 \right\rangle$ have one solution with the irreducible representation $[2]$ in all three channels, and another solution with $[11]$ in all three channels. In the case of $\left\langle V_{(\frac{3}{2},\frac{2}{3})} V_{(\frac{1}{2},0)}^3 \right\rangle$, we can still define a unique solution $F_{[2]}^{(s)}$ with the representation $[2]$ in the $s$-channel, but that solution has both representations $[2]$ and $[11]$ appearing in the $t$ and $u$ channels. In other words, we have nontrivial fusion relations of the type $F_{[2]}^{(s)} = f_{[2],[2]} F_{[2]}^{(t)} + f_{[2],[11]} F_{[11]}^{(t)}$.

The results for $\left\langle V_{(\frac{3}{2},0)} V_{(\frac{1}{2},0)}^3 \right\rangle$ are a strong confirmation that the determination $\Lambda_{(\frac{3}{2},0)} = [3] + [111]$ made in (2.27a) is correct. On the basis of the mere dimensions of representations, one might have proposed the determination $[21] + [1]$ instead, but this would have led us to expect five bootstrap solutions with very different properties.

**When two solutions coincide:** $\left\langle V_{(\frac{3}{2},0)}^2 V_{(\frac{1}{2},0)}^2 \right\rangle$

From $V_{(\frac{3}{2},0)}^{[3]}$ and $V_{(\frac{3}{2},0)}^{[111]}$, we can build 4 four-point functions of the type $\left\langle V_{(\frac{3}{2},0)}^2 V_{(\frac{1}{2},0)}^2 \right\rangle$. However, since $[3] \otimes [111] = [3111] + [411] + [31] + [211]$ and none of these representations appear in the tensor product $[1] \otimes [1]$ (2.13a), we have

$$\left\langle V_{(\frac{3}{2},0)}^{[3]} V_{(\frac{3}{2},0)}^{[111]} V_{(\frac{1}{2},0)}^{[1]} V_{(\frac{1}{2},0)}^{[1]} \right\rangle = \left\langle V_{(\frac{3}{2},0)}^{[111]} V_{(\frac{3}{2},0)}^{[3]} V_{(\frac{1}{2},0)}^{[1]} V_{(\frac{1}{2},0)}^{[1]} \right\rangle = 0 . \tag{4.28}$$

This leaves us with only two non-vanishing correlation functions, for which we introduce $s$-channel bases of solutions:

| 4-point function | $s$-channel solutions | $t, u$-channel representations |
|---|---|---|
| $\left\langle V_{(\frac{3}{2},0)}^{[3]} V_{(\frac{3}{2},0)}^{[3]} V_{(\frac{1}{2},0)}^{[1]} V_{(\frac{1}{2},0)}^{[1]} \right\rangle$ | $F_{[]}^{(s)}, F_{[2]}^{(s)}, F_{[11]}^{(s)}$ | $[4] + [31] + [2]$ |
| $\left\langle V_{(\frac{3}{2},0)}^{[111]} V_{(\frac{3}{2},0)}^{[111]} V_{(\frac{1}{2},0)}^{[1]} V_{(\frac{1}{2},0)}^{[1]} \right\rangle$ | $G_{[]}^{(s)}, G_{[2]}^{(s)}, G_{[11]}^{(s)}$ | $[1111] + [211] + [11]$ |

(4.29)

We numerically find that the 6 solutions are not linearly independent: the space of solutions actually has dimension 5. Nevertheless, it is possible to single out each one of the 6 solutions numerically. The space of solutions for $\left\langle V_{(\frac{3}{2},0)}^{[3]} V_{(\frac{3}{2},0)}^{[3]} V_{(\frac{1}{2},0)}^{[1]} V_{(\frac{1}{2},0)}^{[1]} \right\rangle$ is defined by $D_{(1,1)}^{(t)} = D_{(1,1)}^{(u)} = 0$ (see Eqs. (4.6c), (4.6h) and (4.6i)), and each solution $F_{[]}^{(s)}, F_{[2]}^{(s)}, F_{[11]}^{(s)}$ is singled out by the additional vanishing of two structure constants among $D_1^{(s)}, D_{(1,0)}^{(s)}$, and $D_{(1,1)}^{(s)}$. There is a unique solution $F_0$ such that

$$\mathrm{Span}(F_0) = \mathrm{Span}\left( F_{[]}^{(s)}, F_{[2]}^{(s)} \right) \cap \mathrm{Span}\left( G_{[]}^{(s)}, G_{[2]}^{(s)} \right) , \tag{4.30}$$

which is singled out by requiring $D_{(1,1)}^{(t)} = D_{(1,1)}^{(u)} = D_{(1,0)}^{(t)} = D_{(1,0)}^{(u)} = 0$. We can therefore attribute the existence of only 5 independent solutions to the coincidence of one solution for $\left\langle V_{(\frac{3}{2},0)}^{[3]} V_{(\frac{3}{2},0)}^{[3]} V_{(\frac{1}{2},0)}^{[1]} V_{(\frac{1}{2},0)}^{[1]} \right\rangle$ with one solution for $\left\langle V_{(\frac{3}{2},0)}^{[111]} V_{(\frac{3}{2},0)}^{[111]} V_{(\frac{1}{2},0)}^{[1]} V_{(\frac{1}{2},0)}^{[1]} \right\rangle$.

**When $O(n)$ does not see the difference:** $\left\langle V_{(\frac{3}{2},\frac{2}{3})}^2 V_{(\frac{1}{2},0)}^2 \right\rangle$ **versus** $\left\langle V_{(\frac{3}{2},\frac{2}{3})} V_{(\frac{3}{2},-\frac{2}{3})} V_{(\frac{1}{2},0)}^2 \right\rangle$

The two fields $V_{(\frac{3}{2},\frac{2}{3})}^{[21]}$ and $V_{(\frac{3}{2},-\frac{2}{3})}^{[21]}$ differ by their behaviour with respect to conformal symmetry, but belong to the same $O(n)$ representation. We now study the simplest correlation functions that can see the difference.

We have $\dim \mathrm{Hom}\left([21]\otimes[21],[1]\otimes[1]\right)=5$, with $s$-channel invariants corresponding to the 5 irreducible representations in $2[2]+2[11]+[]$. In the case of $\left\langle V^2_{(\frac{3}{2},\frac{2}{3})}V^2_{(\frac{1}{2},0)}\right\rangle$, we do find 5 solutions of crossing symmetry. But in the case of $\left\langle V_{(\frac{3}{2},\frac{2}{3})}V_{(\frac{3}{2},-\frac{2}{3})}V^2_{(\frac{1}{2},0)}\right\rangle$, we find only 4 solutions.

The difference between the two correlation functions can be attributed to the presence or absence of the identity field in the $s$-channel. In the second case, the identity field is forbidden by the fusion rule (3.4), because $V_{(\frac{3}{2},\frac{2}{3})}\neq V_{(\frac{3}{2},-\frac{2}{3})}$. This implies the constraint $D^{(s)}_1=0$, which is expected to reduce the dimension of the space of solutions by one.

This does not necessarily mean that the identity representation $[]$ of $O(n)$ does not appear in the $s$-channel, though. We may try to define a solution $F^{(s)}_{[]}$ by setting $D^{(s)}_{(1,0)}=D^{(s)}_{(2,1)}=D^{(s)}_{\text{odd spin}}=0$, which would eliminate the $s$-channel representations $[2]$ and $[11]$. However, it turns out that $D^{(s)}_{(2,1)}$ vanishes in all 5 solutions, so setting it to zero is actually not enough for eliminating $[2]$. The appearance of $[]$ is therefore still an open question.

**When solutions are linearly dependent:** $\left\langle V_{(2,0)}V_{(1,0)}V^2_{(\frac{1}{2},0)}\right\rangle$

There are 7 four-point invariants, but only 5 solutions. The representation $\Lambda_{(2,0)}$ (2.27c) is made of 5 irreducible representations, so we have 5 fields of the type $V_{(2,0)}$. In the 4 cases $V^{[4]}_{(2,0)}$, $V^{[22]}_{(2,0)}$, $V^{[211]}_{(2,0)}$, $V^{[]}_{(2,0)}$, there exists a unique corresponding four-point function, which is easy to characterize by the representations that propagate in each channel:

| 4-point function | $s$-channel | $t$-channel | $u$-channel |
|---|---|---|---|
| $\left\langle V^{[4]}_{(2,0)}V^{[2]}_{(1,0)}V^{[1]}_{(\frac{1}{2},0)}V^{[1]}_{(\frac{1}{2},0)}\right\rangle$ | $[2]$ | $[3]$ | $[3]$ |
| $\left\langle V^{[22]}_{(2,0)}V^{[2]}_{(1,0)}V^{[1]}_{(\frac{1}{2},0)}V^{[1]}_{(\frac{1}{2},0)}\right\rangle$ | $[2]$ | $[21]$ | $[21]$ |
| $\left\langle V^{[211]}_{(2,0)}V^{[2]}_{(1,0)}V^{[1]}_{(\frac{1}{2},0)}V^{[1]}_{(\frac{1}{2},0)}\right\rangle$ | $[11]$ | $[21]$ | $[21]$ |
| $\left\langle V^{[2]}_{(2,0)}V^{[2]}_{(1,0)}V^{[1]}_{(\frac{1}{2},0)}V^{[1]}_{(\frac{1}{2},0)}\right\rangle$ | $[2]+[11]+[]$ | $[3]+[21]+[1]$ | $[3]+[21]+[1]$ |
| $\left\langle V^{[]}_{(2,0)}V^{[2]}_{(1,0)}V^{[1]}_{(\frac{1}{2},0)}V^{[1]}_{(\frac{1}{2},0)}\right\rangle$ | $[2]$ | $[1]$ | $[1]$ |

(4.31)

For example, to single out the solution $\left\langle V^{[22]}_{(2,0)}V^{[2]}_{(1,0)}V^{[1]}_{(\frac{1}{2},0)}V^{[1]}_{(\frac{1}{2},0)}\right\rangle$, we require the vanishing of $D^{(s)}_{(1,1)}$, $D^{(s)}_{(2,\frac{1}{2})}$, $D^{(t)}_{(\frac{1}{2},0)}$ and $D^{(t)}_{(\frac{3}{2},0)}$.

The four-point function $\left\langle V^{[2]}_{(2,0)}V^{[2]}_{(1,0)}V^{[1]}_{(\frac{1}{2},0)}V^{[1]}_{(\frac{1}{2},0)}\right\rangle$ a priori corresponds to 3 solutions: $F^{(s)}_{[]}$, $F^{(s)}_{[2]}$ and $F^{(s)}_{[11]}$. The solution $F^{(s)}_{[11]}$ can be determined modulo $\left\langle V^{[211]}_{(2,0)}V^{[2]}_{(1,0)}V^{[1]}_{(\frac{1}{2},0)}V^{[1]}_{(\frac{1}{2},0)}\right\rangle$, by requiring $D^{(s)}_{\text{even spin}}=0$. This leaves us with a 3-dimensional subspace of solutions such that $D^{(s)}_{\text{odd spin}}=0$. We already know 3 such solutions, corresponding to $V^{[4]}_{(2,0)}$, $V^{[22]}_{(2,0)}$, and $V^{[]}_{(2,0)}$. Therefore, the solutions $F^{(s)}_{[]}$ and $F^{(s)}_{[2]}$ are linear combinations of these already determined solutions.

**Nontrivial multiplicity:** $\left\langle V_{(\frac{5}{2},0)} V_{(\frac{1}{2},0)}^3 \right\rangle$

The structure of $\Lambda_{(\frac{5}{2},0)}$ (2.27f) implies the existence of two distinct primary fields of the type $V_{(\frac{5}{2},0)}^{[21]}$. As the tensor product $[1]^{\otimes 3}$ can only reach representations $\lambda$ with $|\lambda| \leq 3$, we therefore have 4 possible four-point functions, one of which comes with a nontrivial multiplicity:

| 4-point function | Multiplicity | $s,t,u$-channel representation |
|---|:---:|:---:|
| $\left\langle V_{(\frac{5}{2},0)}^{[3]} V_{(\frac{1}{2},0)}^{[1]} V_{(\frac{1}{2},0)}^{[1]} V_{(\frac{1}{2},0)}^{[1]} \right\rangle$ | 1 | $[2]$ |
| $\left\langle V_{(\frac{5}{2},0)}^{[111]} V_{(\frac{1}{2},0)}^{[1]} V_{(\frac{1}{2},0)}^{[1]} V_{(\frac{1}{2},0)}^{[1]} \right\rangle$ | 1 | $[11]$ |
| $\left\langle V_{(\frac{5}{2},0)}^{[21]} V_{(\frac{1}{2},0)}^{[1]} V_{(\frac{1}{2},0)}^{[1]} V_{(\frac{1}{2},0)}^{[1]} \right\rangle$ | 2 | $[2]+[11]$ |
| $\left\langle V_{(\frac{5}{2},0)}^{[1]} V_{(\frac{1}{2},0)}^{[1]} V_{(\frac{1}{2},0)}^{[1]} V_{(\frac{1}{2},0)}^{[1]} \right\rangle$ | 1 | $[2]+[11]+[]$ |

(4.32)

The nontrivial multiplicity influences the counting of the 9 four-point invariants: each invariant associated to $\left\langle V^{[21]} V^{[1]} V^{[1]} V^{[1]} \right\rangle$ has to be counted twice, for a total of 4 invariants.

Numerically, we find 6 independent solutions of crossing symmetry equations. The solutions associated to $\left\langle V_{(\frac{5}{2},0)}^{[3]} V_{(\frac{1}{2},0)}^{[1]} V_{(\frac{1}{2},0)}^{[1]} V_{(\frac{1}{2},0)}^{[1]} \right\rangle$ and $\left\langle V_{(\frac{5}{2},0)}^{[111]} V_{(\frac{1}{2},0)}^{[1]} V_{(\frac{1}{2},0)}^{[1]} V_{(\frac{1}{2},0)}^{[1]} \right\rangle$ are easily singled out by allowing only one representation to propagate in each channel. The remaining 4 solutions are harder to disentangle, and we cannot say whether the two copies of $V_{(\frac{5}{2},0)}^{[21]}$ lead to linearly independent solutions.

# 5 Conclusion and outlook

## 5.1 Solving the $O(n)$ CFT: achievements and challenges

We have completed the determination of the spectrum of the $O(n)$ CFT, and we now know the list of primary fields, together with their behaviour under the global $O(n)$ symmetry. In order to solve the $O(n)$ CFT, it remains to compute the correlation functions of these fields. We have done this numerically in a number of examples, and inferred some exact results about fusion rules and numbers of solutions of crossing symmetry equations. Let us discuss the interpretation of our results, and outline what remains to be done.

**Coincidences of solutions of crossing symmetry**

Given a four-point function, we have argued that the number $\mathcal{N}$ of independent solutions of crossing symmetry cannot exceed the number $\mathcal{I}$ of four-point $O(n)$ invariants. All four-point functions of the fields $V_{(\frac{1}{2},0)}$, $V_{(1,0)}$ and $V_{(1,1)}$ actually obey $\mathcal{N} = \mathcal{I}$. As the four-point functions become more complicated, we find examples where $\mathcal{N} < \mathcal{I}$, and it might well be that $\mathcal{N} = \mathcal{I}$ occurs only in a finite number of cases. There are two plausible interpretations for the existence of cases where $\mathcal{N} < \mathcal{I}$:

- *These may reflect the existence of a larger symmetry.* If conformal symmetry and $O(n)$ symmetry do not explain all relations between solutions of crossing symmetry, is there a larger symmetry at work? In the spectrum (2.20), the $O(n)$ representations $\Lambda_{(r,s)}$ are not irreducible, but they are sums of (typically many) irreducible representations. There are

indications that the representations $\Lambda_{(r,s)}$ follow from an algebra that is larger than $O(n)$ [41]. However, while it does act on the spectrum, this algebra is not a symmetry algebra of the model, because the representations $\Lambda_{(r,s)}$ do not close under tensor products.

More generally, the linear relations that lead to $\mathcal{N} < \mathcal{I}$ cannot be explained by grouping $O(n)$ representations into larger representations, be they $\Lambda_{(r,s)}$ or smaller representations: if there is a larger symmetry, it must work in a more subtle way.

- *These may be dynamical accidents.* Until we find an explanation from symmetry, we have to consider coincidences of solutions of crossing symmetry as dynamical accidents. For example, we saw that each one of the two four-point functions of Table (4.29) gives rise to three well-identified solutions of crossing symmetry. It just happens that these six solutions are linearly dependent, so that $5 = \mathcal{N} < \mathcal{I} = 6$ for the four-point function $\left\langle V_{(\frac{3}{2},0)}^2 V_{(\frac{1}{2},0)}^2 \right\rangle$.

  Coincidences of solutions of crossing symmetry are probably comparable to coincidences of dimensions of primary fields, starting with the two fields $V_{(\frac{3}{2},0)}^{[3]}$ and $V_{(\frac{3}{2},0)}^{[111]}$ which belong to different $O(n)$ representations but have the same conformal dimensions. These coincidences are probably specific to the two-dimensional $O(n)$ CFT: in higher dimensions, we would expect $\mathcal{N} = \mathcal{I}$ in all cases.

The number of invariants $\mathcal{I}$ is determined by the action of $O(n)$ on the spectrum, and therefore by our conjecture (2.26) for $\Lambda_{(r,s)}$. As we have seen in Section 4.3, numerical bootstrap results provide strong evidence for our conjecture.

### Fusion rules and field multiplicities

In Sections 4.1 and 4.2, we have determined the fusion products of the fields $V_{(\frac{1}{2},0)}$, $V_{(1,0)}$ and $V_{(1,1)}$ with one another. Our results are given by Eqs. (4.16), (4.20)–(4.21), (4.23) and (4.25)–(4.26).

These products take into account $O(n)$ symmetry, and are thereby finer than the Virasoro-only fusion products that were recently conjectured by Ikhlef and Morin-Duchesne based on lattice arguments in related CFTs [42]. The Virasoro-only products obey the non-trivial rule $V_{(r,s)} \in V_{(r_1,s_1)} \times V_{(r_2,s_2)} \implies r \geq |r_1 - r_2|$. We found counter-examples to this rule, such as the field $V_{(1,1)}$ propagating in $\left\langle V_{(\frac{5}{2},0)} V_{(\frac{1}{2},0)}^3 \right\rangle$. Maybe the rule's applicability depends on the behaviour of the involved fields under $O(n)$ transformations.

However, the fusion rules that we have established do not take into account field multiplicities, i.e., they do not distinguish two fields that transform similarly under the conformal and $O(n)$ symmetries. For example, according to $\Lambda_{(3,0)}$ (2.27h), there are four different fields of the type $V_{(3,0)}^{[22]}$. Taming field multiplicities will be needed for writing complete fusion rules, and therefore for solving the CFT. In the given example, knowing only that *some* fields of the type $V_{(3,0)}^{[22]}$ appear in two fusion rules $V_1 \times V_2$ and $V_3 \times V_4$, we cannot deduce that $V_{(3,0)}^{[22]}$ appears in the $s$-channel decomposition of the four-point function $\left\langle V_1 V_2 V_3 V_4 \right\rangle$. To deduce that, we would need to know precisely *which* fields of the type $V_{(3,0)}^{[22]}$ appear, and specifically that at least one of those fields is common to the two fusion rules.

### Structure constants and nonlinear bootstrap

We have been treating crossing symmetry as a system of linear equations (3.16) for the four-point structure constants $D_{(r,s)}$. However, an essential axiom of the conformal bootstrap is that each four-point structure constant is the product of two three-point structure constants. In

terms of three-point structure constants, crossing symmetry is a system of quadratic equations, and it is much more constraining, because there are much fewer three-point structure constants than four-point structure constants.

Our notations for four-point structure constants $D_{(r,s)}$ may be misleading, because they only indicate the dependence on conformal dimensions, and not on $O(n)$ representations. In fact, all structure constants of the CFT depend on $O(n)$ representations, just like the fields $V_{(r,s)}^{\lambda}$. For example, the $s$-channel four-point structure constants for the four-point function $\left\langle V_{(\frac{1}{2},0)}^{4} \right\rangle$ depend on $\lambda \in \left\{ [\,], [2], [11] \right\}$, via their dependence on the choice of an $s$-channel solution $F_{\lambda}^{(s)}$.

Dealing with nonlinear bootstrap equations is surely the next technical step in the study of the $O(n)$ CFT and cognate CFTs. Of course, we could take the four-point structure constants as determined by our current method, and deduce three-point structure constants. This would provide strong tests of the consistency of our results, as the same three-point structure constant may appear in several different four-point structure constants. The question is whether we could do better, and simultaneously solve crossing symmetry equations for several four-point functions that share some three-point structure constants.

There is also the issue of finding analytic formulas for structure constants. In the critical $Q$-state Potts model, some structure constants are known analytically: to begin with, the simplest three-point structure constant is given by the Delfino–Viti conjecture [43], which has been numerically tested to high precision [32]. Moreover, some ratios between structure constants have been determined analytically (in the form of ratios of integer-coefficient polynomials in $Q$), both from the lattice model [30] and from the numerical bootstrap [32]. In the $O(n)$ CFT, we have been able to determine some ratios of structure constants analytically, but so far we have found no analogue of the Delfino–Viti conjecture.

## 5.2 Geometrical interpretation of the CFT

In the lattice description, the partition function of the $O(n)$ model is a sum over loops on a two-dimensional lattice. We will now discuss how this may lead to a geometrical interpretation of correlation functions in the $O(n)$ CFT. The ultimate aim is to determine fusion rules and numbers of crossing symmetry solutions, by counting two-dimensional topological graphs. We do not understand this systematically at the moment, so we will only sketch a few ideas.

### $O(n)$ representations and watermelon operators

The basic spin observable $\phi(x)$ in the lattice model transforms in the vector representation of $O(n)$, and its continuum limit is the field $V_{(\frac{1}{2},0)}$ in the $O(n)$ CFT [29]. Using the high-temperature expansion, correlation functions of $\phi(x)$ can be represented as sums over lines and loops. For example, the two-point function $\left\langle \phi(x)\phi(y) \right\rangle$ is a sum over graphs where, in addition to the loops that already appear in the partition function, there is a line connecting $x$ and $y$.

In the continuum limit, this suggests that the field $V_{(\frac{1}{2},0)}(z)$ inserts one line at point $z$. In a correlation function, this line has to end at the position of another field. For example, in the four-point function $\left\langle V_{(\frac{1}{2},0)}^{4} \right\rangle$, there are three ways to connect the four fields, see Figure (4.8). The lines in that figure originally indicated contractions of $O(n)$ vector indices: this is an early hint that the model's geometrical and algebraic interpretations are closely related.

Similarly, the field $V_{(r,0)}^{[2r]}$ can be thought of as a $2r$-leg operator, i.e., an operator that inserts

$2r$ lines. Such operators are called watermelon or fuseau operators [5, 44]:

$$\left\langle V_{(3,0)}^{[6]} V_{(3,0)}^{[6]} \right\rangle \qquad \longrightarrow \qquad \tag{5.1}$$

In the lattice model, a $2r$-leg operator is built from $2r$ spin operators $\phi(x)$ inserted at neighboring sites, with their $O(n)$ labels projected onto the traceless symmetric tensor representation $[2r]$. In the continuum limit, the neighboring sites coincide. In correlation functions, due to the tracelessness of the representation $[2r]$, two legs of the same operator cannot be connected to one another, but have to be connected to legs of other operators.

More generally, the field $V_{(r,s)}(z)$ with $s \neq 0$ can be interpreted as an operator that inserts $2r$ lines, with the rule that a line that winds around its position $z$ picks a phase $e^{i\pi s}$. (This means that the weight of a configuration with such a winding line is multiplied by $e^{i\pi s}$ [1, 44].) Actually, the notion of winding around an operator already makes sense on the lattice, even though the lattice operator is built from $2r$ distinct sites: since these sites are neighbors, a line cannot wind around only a subset of these sites. Since the conformal spin $rs$ is integer, the phase factor is trivial if all $2r$ lines wind around $z$, and this allows watermelon operators to be mutually local. To see that the phase factors allow us to distinguish different values of $s$, consider the two-point functions of $V_{(1,0)}$ and $V_{(1,1)}$. In both cases, there are two inequivalent graphs:

$$\left\langle V_{(1,0)} V_{(1,0)} \right\rangle \text{ or } \left\langle V_{(1,1)} V_{(1,1)} \right\rangle \qquad \longrightarrow \qquad \tag{5.2}$$

The bottom graph comes with a minus sign in the case of $\left\langle V_{(1,1)} V_{(1,1)} \right\rangle$, which therefore differs from $\left\langle V_{(1,0)} V_{(1,0)} \right\rangle$.

**Young projectors and line crossings**

Phase factors are enough for singling out fully symmetric or antisymmetric representations $[2r]$ or $[1^{2r}]$. However, in general they cannot distinguish the various fields $V_{(r,s)}^{\lambda}$ that have the same conformal dimensions, but belong to different $O(n)$ representations. To single out a representation $\lambda$, we should also apply a Young projector $P_{\lambda}^{2r} : [1]^{2r} \to \lambda$,

$$\tag{5.3}$$

We do not know how to do this systematically, because lines are not allowed to cross in our graphs.

Young projectors are algebraic objects, which a priori know nothing about our two-dimensional model and the topology of our two-dimensional graphs. If we identified each line with a vector representation and applied Young projectors, we would sum over all permutations of

lines, and obtain graphs with line crossings. In two dimensions, if we do not want lines to cross, only cyclic permutations are allowed.

A cyclic permutation of $2r$ lines has the signature $(-1)^{2r+1}$: it is odd for two lines, even for three lines, etc. As a consequence, the antisymmetrizer $P^3_{[111]}$ becomes trivial if we only sum over cyclic permutations. This explains why the representations $[3]$ and $[111]$ can both appear in $\Lambda_{(\frac{3}{2},0)}$ (2.27a), and lead to fields $V^{[3]}_{(\frac{3}{2},0)}$ and $V^{[111]}_{(\frac{3}{2},0)}$ with the same conformal dimensions and two-point functions.

This however does not imply that $V^{[3]}_{(\frac{3}{2},0)}$ and $V^{[111]}_{(\frac{3}{2},0)}$ coincide. This is because in higher correlation functions, the Young projectors associated to fields at different positions are expected to mix. For example, in the context of the four-point functions of Eq. (4.29), we do see a difference between these two fields.

Similarly, after projection on the representation $[21]$, cyclic permutations of three lines have the eigenvalues $e^{\pm\frac{2\pi i}{3}}$, hence $[21]$ appears in $\Lambda_{(\frac{3}{2},\frac{2}{3})}$.

**Algebraic interpretation**

The foregoing analysis, should be greatly facilitated by turning to a more algebraic description, the principles of which we briefly sketch now. Graphs with crossings can be conveniently described using the formalism of diagram algebras—in this case, the Brauer algebra [45]. Technically, the Brauer algebra is the Schur–Weyl dual of $O(n)$ in the tensor product of vector representations: this implies that irreducible representations of the Brauer algebra are labelled by Young diagrams, just like irreducible representations of $O(n)$. The size $|\lambda|$ of a Young diagram corresponds to the number of legs of a site in the graph.

Forbidding crossings amounts to restricting to a subalgebra of the Brauer algebra, called the Jones–Temperley–Lieb algebra (JTL) [23, 46]. Remarkably, we have found in preliminary investigations that when we decompose irreducible representations of the Brauer algebra into direct sums of irreducible JTL representations, the number of legs can increase from $|\lambda|$ to $|\lambda| + 2n$ with $n \in \mathbb{N}$. This is at the root of the appearance of $O(n)$ representations with various sizes $|\lambda| \leq 2r$ in $\Lambda_{(r,s)}$ (2.26).

We also note that in the continuum limit, the JTL algebra becomes the interchiral algebra [40]: a symmetry algebra of the CFT that includes conformal symmetry.

## 5.3 Analytic continuation of the $O(n)$ model

As we have argued in the introduction, the $O(n)$ CFT is defined on infinitely many copies of the complex $n$-plane. Beyond $n \in [-2, 2]$, does the $O(n)$ CFT still describe the critical point of some $O(n)$ lattice model or field theory?

From a formal point of view, the critical coupling $K_c(n)$ (1.1) has two branch cuts $\pm(2, \infty)$, which correspond to vertical lines in the $\beta^2$-plane (1.5). It is tempting to conjecture that the model is critical for $n \in \mathbb{C} - ((2, \infty) \cup (-\infty, -2))$. If the situation is the same as in the CFT, we should still be able to analytically continue through the branch cuts, but the result would depend on whether we come from above or below.

In order to find out, we would need to carry out extensive numerical simulations of the lattice model with complex $n$ and coupling constants. For the moment, let us collect some preliminary remarks on the question.

**The real axis**

The lattice $O(n)$ model that we have discussed in the introduction is naturally defined for $n \in [-2, 2]$. For $n > 2$, the lattice model (and all the refinements that have been studied

so far) does not seem to have a phase transition at finite, real temperature, and cannot be related to the $O(n)$ CFT. This state of affairs is, for $n$ integer at least, a consequence of the Mermin–Wagner theorem which precludes spontaneous breaking of continuous symmetries in two dimensions [47]—although this theorem leaves open the possibility of having Kosterlitz–Thouless phases like for $n = 2$ [48]. Note how the situation is different from the case of the $Q$-state Potts model, which retains a critical point for $Q > 4$, albeit a first-order critical point.

This can be made clearer using the Lagrangian description of the $O(n)$ model as a non-linear sigma model whose target space is the sphere $S^{n-1}$. In this model, the $n$-component vector $\phi$ subject to $\phi \cdot \phi = 1$ is no longer defined on a lattice, but on a continuous two-dimensional space. The Lagrangian density is

$$\mathcal{L} = \frac{1}{2g^2} \partial_\mu \phi \cdot \partial_\mu \phi \ . \tag{5.4}$$

The beta function for the coupling is

$$\beta(g^2) = (n-2)g^4 + O(g^6) \ . \tag{5.5}$$

For $n > 2$, the renormalization group flow is towards strong coupling at large length scales, and it is expected that the symmetry is restored and no transition occurs (that is the Mermin–Wagner theorem). For $n < 2$, on the other hand, zero coupling is an attractive fixed point, corresponding to a spontaneously broken symmetry phase. It is believed that for $n \geq -2$ there is a second-order phase transition separating this regime from the strong coupling regime, and that the dilute $O(n)$ CFT describes the corresponding critical point.

Strictly speaking, the only physical value of $n < 2$ is $n = 1$, which corresponds to the Ising universality class. To consider real values of $n < 2$ requires performing an analytic continuation of perturbative results at $n \in \mathbb{N}^*$. The case $n = 2$ is a bit special, as the beta function vanishes perturbatively to all orders. The model then exhibits a line of fixed points with continuously varying scaling dimensions. The corresponding critical phase terminates at the Kosterlitz–Thouless point, which is described by the free boson CFT.

The physics encompassed in the beta function (5.5) do not seem to be affected by the bound $n = -2$, and the critical coupling $K_c(n)$ admits a real determination for $n < -2$. This seems to suggest that the sigma model still exhibits a phase transition that may be described by the $O(n)$ CFT. In fact, there is evidence that the model for $n < -2$ does have a phase transition, albeit a first-order one, and that this transition is closely related with the first-order phase transition in the Potts model at $Q > 4$ [1].

Therefore, the sigma model picture confirms that the $O(n)$ CFT is not the critical limit of the $O(n)$ model for $n \in (-\infty, -2) \cup (2, \infty)$.

**Complex values of $n$**

For $\Re n < 2$, the behaviour of the perturbative beta function (5.5) is not qualitatively affected near the line $g^2 \in \mathbb{R}$ by allowing $n$ to be complex. This suggests that the $O(n)$ CFT may be related to the lattice model and the sigma model. This might even hold for $\Re n < -2$, as the first-order phase transition may morph into a second-order phase transition as $n$ becomes complex.

For $\Re n > 2$, the small-coupling phase definitely seems unstable, just like for $n \in (2, \infty)$, where the model is expected not to have a phase transition. For the CFT to describe the critical behavior of the $O(n)$ lattice model, something has to happen: this might involve higher-order terms in the beta function, or the emergence of relevant operators not taken into account in the Lagrangian (5.4).

An example of the latter situation is well-known to occur in the case $n = 2$. The free boson CFT makes sense for all values $R \in \mathbb{C}^*$ of the compactification radius, but only describes the

continuum limit of the lattice XY model in the Kosterlitz–Thouless (KT) phase, $0 < R < \sqrt{2}$. (In our convention, $R = 0$ is the low-temperature limit, $R = \frac{1}{\sqrt{2}}$ the self-dual point, and $R = \sqrt{2}$ the critical temperature or KT point.) This is because the lattice model is bound to contain local vortices, which are absent from the free boson action, since they can be described by perturbations that are irrelevant in the KT phase. However, for $R \geq \sqrt{2}$, these perturbations become relevant, and draw the XY model into a massive phase. This is not seen in the beta function (5.5), which vanishes to all orders for $n = 2$.

In our case, we would need terms such as vortices to render the system critical, instead of destroying criticality. This may be less unlikely than it seems, since we now allow complex couplings. There are indeed known examples where relevant perturbations with complex couplings give rise to flows towards other critical points [49]. Whether this happens in our case is not known.

## Acknowledgements

We are grateful to Shiliang Gao, Slava Rychkov, Bernardo Zan, and Jean-Bernard Zuber, for helpful discussions or correspondence. We would like to thank the organizers and participants of Bootstat 2021 for a stimulating program. We are grateful to Bernardo Zan and Ioannis Tsiares for comments on the draft of this article. We would like to thank the anonymous SciPost reviewers for their helpful suggestions.

This paper is partly a result of the ERC-SyG project, Recursive and Exact New Quantum Theory (ReNewQuantum) which received funding from the European Research Council (ERC) under the European Union's Horizon 2020 research and innovation programme under grant agreement No 810573. The work was also supported by the ERC NuQFT Project, which received similar funding under grant agreement No 669205.

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
