# Peer review of "Global symmetry and conformal bootstrap in the two-dimensional $O(n)$ model"

_SciPost Physics, doi:SciPost Phys. 12, 147 (2022)_

## Round 2 · Referee Report · Anonymous (Referee 1) · 2022-2-13

Strengths

  1. Very thorough paper
  2. The method devised by the authors is applied to many correlation functions, with interesting results

Weaknesses

  1. Sometimes the exposition appears to be overcomplicated

Report

The authors fix the conformal data of the two dimensional $O(n)$ CFT by studying four point functions of (many) lower dimension operators; they are able to do this at generic value of $n$, and obtain very good numerical results. This fixes previously unknown conformal data of the theory.

The paper has a lot of detailed explanations of the method used by the authors, the method is interesting, the authors point out valid future directions of work; it's a very good paper and I recommend publication, after very minor changes. Let me point out that I believe sometimes the exposition of some topics could be simplified by a lot, making it more accessible to the average reader (an example would be the "Four point $O(n)$ invariants" section, in my opinion).

Requested changes

  1. It appears that linear combinations of irreps are taken with constant ($n$ independent) coefficients. If this is the case, it should be mentioned explicitly above (2.21); at the moment it's only mentioned that these coefficients are positive and integer, but it's not excluded that they could depend on $n$.

  2. I find the discussion between equations (3.35) and (3.39) to be a bit detached from the rest of the paper, and I personally don't really understand it. It would be good to explain better what the vector $d$ and the matrices $b$ are. For example, how are the $d^{(s)}$ related to $D^{(s)}_V$ in eq (3.16) ? What are the matrices $b$? Some more explanation would be good.

  3. In Table (3.42), it's unclear to me why operator $(1,0)$ and $(1,1)$ are those that get removed, and not the other ones. Is it an arbitrary choice, in order to find one of the possible solutions, or is there another reason?

  4. In (4.5), where does the conjecture that $F_{\lambda}$ is finite come from? It appears to me that it follows from the decomposition of irreps in (2.27) (correct me if I'm wrong), but since this has been carried out only for finitely many operators, how can we know that we need to subtract only finitely many operators from these spectra? I doubt this would affect the numerics, since we're truncating the number of operators, but it would still be good to add some details to this conjecture.

---

## Round 2 · Referee Report · Anonymous (Referee 2) · 2022-3-3

Report

A conformal field theory (CFT) is considered as solved if its operator content and all the structure constants (coefficients of the operator product expansion) are known, since this knowledge allows in principle to express all the correlation functions. This program was originally completed in the '80s for minimal models of 2D CFT exploiting the fact that they contain only degenerate primary operators. The differential equations they provide allow to determine the spectrum of conformal dimensions and the four-point functions, from which the structure constants can be obtained by fusion.

The present paper is devoted to the CFT of the critical 2D O(n) model with values of n varying continuously between -2 and 2. The possibility of the continuation to noninteger n is known, as well as the fact that it no longer allows to rely on differential equations for the correlation functions of the fundamental operators, which become nondegenerate. The alternative strategy adopted by the authors proceeds through two main steps. They first conjecture the operator content starting from the known continuum limit of the partition function on the honeycomb lattice. The conjecture is based on a very detailed analysis which allows the authors to propose a classification of the operators according to their transfomation properties under both conformal and O(n) symmetries. The second main step is to use this information to organize the bootstrap of four-point functions according to the representations that can propagate in the different fusion channels. The crossing equations are then solved numerically imposing a cutoff on conformal dimensions and checking the stability of the solutions against increase of the cutoff. The numerical results are convincing and support the assumptions made along the way. The authors analyze the solutions of the crossing equations for several nondegenerate operators, argue an upper bound on the number of solutions, and deduce fusion rules. The many subtleties of the analysis are discussed and illustrated through examples. The crossing equations are numerically solved as linear equations for the coefficients of the decomposition over the operators propagating in fusion channels, but solving for the structure constants themselves appears as a realistic target for the future. The authors observe that it will also be interesting to look for analytic formulas for structure constants, as Delfino and Viti showed that this is possible for the related Q-state Potts model.

I recommend publication in SciPost.

---

## Round 3 · Referee Report · Anonymous (Referee 1) · 2022-3-26

Report

I am happy with the changes and recommend publication.

---

## Round 3 · Author Response

We are grateful to the Editor and reviewers for their work, and in particular to the author of Report 1 for the helpful suggestions. Following the four numbered suggestions, we have made the following changes:

  1. We have stated explicitly that the coefficients are $n$-independent.

  2. We have rewritten that discussion in order to clarify it and make the matrices more explicit.

  3. After Table (3.43), we now state that the excluded field may be chosen arbitrarily, and that we chose them by increasing values of the conformal dimension.

  4. The conjecture is now displayed as Eq. (4.5), and the argument for the conjecture is more detailed, including the new Table (4.6).

Moreover, Report 1 suggests that the paragraph on "Four-point $O(n)$ invariants" could be simplified. We do not see how to simplify it while keeping the needed results like Eq. (3.26). What we have done is to add a review of the notation Hom and its properties, see Eq. (3.20) and the preceding text. We hope that this clarifies the paragraph.

---

## Editorial Decision

published